# Persuading Farsighted Receivers in MDPs: the Power of Honesty

**Martino Bernasconi**[*]
Bocconi University
martino.bernasconi@unibocconi.it

**Matteo Castiglioni**
Politecnico di Milano
matteo.castiglioni@polimi.it

**Alberto Marchesi**
Politecnico di Milano
alberto.marchesi@polimi.it

**Mirco Mutti**[*]
Technion
mirco.m@technion.ac.il

## Abstract

*Bayesian persuasion* studies the problem faced by an informed sender who strategically discloses information to influence the behavior of an uninformed receiver. Recently, a growing attention has been devoted to settings where the sender and the receiver interact *sequentially*, in which the receiver's decision-making problem is usually modeled as a *Markov decision process* (MDP). However, previous works focused on computing optimal information-revelation policies (a.k.a. *signaling schemes*) under the restrictive assumption that the receiver acts *myopically*, selecting actions to maximize the one-step reward and disregarding future rewards. This is justified by the fact that, when the receiver is *farsighted* and thus considers future rewards, finding an optimal Markovian signaling scheme is NP-hard. In this paper, we show that Markovian signaling schemes do *not* constitute the "right" class of policies. Indeed, differently from most of the MDPs settings, we prove that Markovian signaling schemes are *not* optimal, and general *history-dependent* signaling schemes should be considered. Moreover, we also show that history-dependent signaling schemes circumvent the negative complexity results affecting Markovian signaling schemes. Formally, we design an algorithm that computes an optimal and $\epsilon$-persuasive history-dependent signaling scheme in time polynomial in $1/\epsilon$ and in the instance size. The crucial challenge is that general history-dependent signaling schemes cannot be represented in polynomial space. Nevertheless, we introduce a convenient subclass of history-dependent signaling schemes, called *promise-form*, which are as powerful as general history-dependent ones and efficiently representable. Intuitively, promise-form signaling schemes compactly encode histories in the form of *honest* promises on future receiver's rewards.

## 1 Introduction

*Bayesian persuasion* [Kamenica and Gentzkow, 2011] is the problem faced by an informed *sender* who wants to influence the behavior of an uninformed, self-interested *receiver* through the provision of payoff relevant information. Bayesian persuasion captures many fundamental problems arising in real-world applications, *e.g.*, online advertising [Bro Miltersen and Sheffet, 2012], voting [Alonso and Câmara, 2016, Castiglioni et al., 2020a, Castiglioni and Gatti, 2021], traffic routing [Bhaskar et al., 2016, Castiglioni et al., 2021a], recommendation systems [Mansour et al., 2016], security [Rabinovich et al., 2015, Xu et al., 2016], marketing [Babichenko and Barman, 2017, Candogan, 2019], medical research [Kolotilin, 2015], and financial regulation [Goldstein and Leitner, 2018].

---

[*]Work done while the author was at Politecnico di Milano.

37th Conference on Neural Information Processing Systems (NeurIPS 2023).

Most of the previous works study the classical, one-shot version of the Bayesian persuasion problem. However, in many application scenarios it is natural to assume that the sender and the receiver interact multiple times in a sequential manner. In spite of this, only a few very recent works addressed *sequential* versions of the Bayesian persuasion problem [Wu et al., 2022, Bernasconi et al., 2022, Gan et al., 2022a,b]. In particular, Wu et al. [2022] and Gan et al. [2022a,b] study settings where the sender and the receiver interact sequentially in a *Markov decision process* (MDP).

In Bayesian persuasion problems in MDPs, at each step of the interaction both the sender and the receiver know the current state of the MDP, and the former has also access to some *private observation* drawn according to a commonly-known, state-dependent distribution. The sender commits beforehand to an information-revelation policy, which is implemented by means of a *signaling scheme* that sends randomized *action recommendations* to the receiver, conditioned on the current (public) state and the sender's private observation. Specifically, the sender commits to a *persuasive* signaling scheme, meaning that the receiver is always incentivized to follow recommendations. At the end of each step, the next state of the MDP and the agents' rewards are determined as a function of the current state, the action actually played by the receiver, and the sender's private observation in the current step.

Wu et al. [2022] and Gan et al. [2022a,b] provide algorithms that compute an optimal (*i.e.*, reward-maximizing) persuasive signaling scheme under the restrictive assumption that the receiver acts *myopically*, selecting actions to maximize the one-step reward and disregarding future ones. This is justified by the fact that, when the receiver is *farsighted* and thus considers future rewards, finding an optimal Markovian signaling scheme is NP-hard to approximate, as shown by Gan et al. [2022a] for infinite-horizon MDPs. An analogous result also holds in finite-horizon MDPs for non-stationary Markovian signaling schemes, as we prove in this work as a preliminary result.

In this paper, we show that Wu et al. [2022] and Gan et al. [2022a,b] failed to provide positive results with farsighted receivers since Markovian signaling schemes do not constitute the "right" class of policies to consider. This is in stark contrast with most of the MDPs settings in which Markovian policies are optimal. Indeed, we prove that Markovian signaling schemes are *not* optimal, and general *history-dependent* signaling schemes should be considered. As a result, we focus on the problem of computing an optimal persuasive history-dependent signaling scheme. Surprisingly, we show that taking history into account allows to circumvent the negative result affecting Markovian signaling schemes. We do that by providing an approximation scheme that finds an optimal $\epsilon$-persuasive (*i.e.*, one approximately incentivizing the receiver to follow action recommendations) history-dependent signaling scheme in time polynomial in $1/\epsilon$ and the size of the problem instance.

The crucial challenge in designing our approximation scheme is that general, history-dependent signaling schemes cannot be represented in polynomial space. Our algorithm overcomes such an issue by using a convenient subclass of history-dependent signaling schemes, which we call *promise-form* signaling schemes. The core idea of such signaling schemes is to compactly encode all the relevant information contained in an history into a *promise* on future receiver's rewards. At each step of the process, a promise-form signaling scheme does *not* only determines an action recommendation for the receiver, but it also makes a promise to them. First, we prove that promise-form signaling schemes are as powerful as general history-dependent ones. Then, we show how an optimal $\epsilon$-persuasive promise-form signaling scheme can be computed in polynomial time by means of a recursive procedure. To do that, we rely on a crucial result showing that, for signaling schemes that *honestly* keep promises made to the receiver, persuasiveness constraints can be expressed as conditions defined locally at each step of the MDP, since the receiver only cares about sender's promises on their future rewards.[2]

## 2  Preliminaries

In this work, we study Bayesian persuasion problems where a *farsighted* receiver takes actions in a *time-inhomogeneous finite-horizon* MDP [Puterman, 2014]. Formally, a problem instance is a tuple

$$\left( \mathcal{S}, \mathcal{A}, \mathcal{H}, \Theta, \{r_h^{\mathsf{S}}\}_{h \in \mathcal{H}}, \{r_h^{\mathsf{R}}\}_{h \in \mathcal{H}}, \{p_h\}_{h \in \mathcal{H}}, \{\mu_h\}_{h \in \mathcal{H}}, \beta \right), \text{ where:}$$

$\mathcal{S}$ is a finite set of states, $\mathcal{A}$ is a finite set of receiver's actions available in each state, $\mathcal{H} := [1, \ldots, H]$ is a set of time steps with $H$ being the time horizon, $\Theta$ is a finite set of sender's private observations, $r_h^{\mathsf{S}}, r_h^{\mathsf{R}} : \mathcal{S} \times \mathcal{A} \times \Theta \to [0, 1]$ are reward functions for the sender and the receiver, respectively,

---

[2]The complete proofs of all the results in the paper can be found in Appendices C, D, and E.

$p_h : \mathcal{S} \times \mathcal{A} \times \Theta \to \Delta(\mathcal{S})$ is a transition function, $\mu_h : \mathcal{S} \to \Delta(\Theta)$ is a function defining probabilities of sender's private observations at each state, while $\beta \in \Delta(\mathcal{S})$ is the initial state distribution.[3]

We consider the most general setting in which the sender commits to a *non-stationary* and *non-Markovian signaling scheme* (henceforth called *history-dependent* signaling scheme for short). For every step $h \in \mathcal{H}$ and state $s_h \in \mathcal{S}$ reached at that step, a history-dependent signaling scheme defines a randomized mapping from sender's private observations to action recommendations for the receiver, based on the whole history of states and receiver's actions observed up to step $h$.[4] Formally, in the following we let $\mathcal{T}_h$ be the set of all the possible *histories* up to step $h$, which is defined as

$$\mathcal{T}_h := \{\tau \mid \tau = (s_1, a_1, \ldots, s_{h-1}, a_{h-1}, s_h) : s_i \in \mathcal{S}, a_i \in \mathcal{A}\},$$

while we let $\mathcal{T} := \mathcal{T}_1 \cup \ldots \cup \mathcal{T}_H$ be the set of all the possible histories (of any length).[5] Then, a history-dependent signaling scheme is defined as a set $\phi := \{\phi_\tau\}_{\tau \in \mathcal{T}}$ of functions $\phi_\tau : \Theta \to \Delta(\mathcal{A})$, which define a mapping from sender's private observations to probability distributions over action recommendations for every possible history.

The interaction between the sender and the receiver goes as follows (Algorithm 1). **(i)** The sender publicly commits to a history-dependent signaling scheme $\phi := \{\phi_\tau\}_{\tau \in \mathcal{T}}$. **(ii)** An initial state $s_1 \sim \beta$ is drawn. **(iii)** At each step $h \in \mathcal{H}$, both agents observe the current state $s_h \in \mathcal{S}$ and the sender also gets a private observation $\theta_h \in \Theta$ drawn according to $\mu_h(s_h)$, with the function $\mu_h$ being known to both the sender and the receiver. **(iv)** The sender communicates to the receiver an action recommendation $a_h \in \mathcal{A}$ sampled according to $\phi_{\tau_h}(\theta_h)$, where $\tau_h \in \mathcal{T}_h$ is the history up to step $h$. **(v)** The receiver plays an action $\widehat{a}_h \in \mathcal{A}$, possibly different from $a_h$. **(vi)** The sender and the receiver get rewards $r_h^{\mathsf{R}}(s_h, a_h, \theta_h)$ and $r_h^{\mathsf{S}}(s_h, a_h, \theta_h)$, respectively. **(vii)** If $h = H$, the interaction ends, otherwise

---

**Algorithm 1** Sender-receiver interaction

1: Sender publicly commits to $\phi := \{\phi_\tau\}_{\tau \in \mathcal{T}}$
2: $s_1 \sim \beta, \tau_1 \leftarrow (s_1), deviated \leftarrow \texttt{False}$
3: **for** each step $h = 1, \ldots, H$ **do**
4:     Sender observes $\theta_h \sim \mu_h(s_h)$
5:     **if** $deviated$ is $\texttt{False}$ **then**
6:         Sender recommends $a_h \sim \phi_{\tau_h}(\theta_h)$
7:         Receiver plays $\widehat{a}_h \in \mathcal{A}$
8:     **else**
9:         Receiver plays $\widehat{a}_h \in \mathcal{A}$
10:     Sender collects reward $r_h^{\mathsf{S}}(s_h, \widehat{a}_h, \theta_h)$
11:     Receiver collects reward $r_h^{\mathsf{R}}(s_h, \widehat{a}_h, \theta_h)$
12:     Next state: $s_{h+1} \sim p_h(s_h, \widehat{a}_h, \theta_h)$
13:     Update history: $\tau_{h+1} \leftarrow \tau_h \oplus (\widehat{a}_h, s_{h+1})$
14:     **if** $a_h \neq \widehat{a}_h$ **then**
15:         $deviated \leftarrow \texttt{True}$

---

the next state $s_{h+1} \sim p_h(s_h, \widehat{a}_h, \theta_h)$ is drawn and the interaction continues to step $h + 1$ starting from the third point. As customary in the literature (see, *e.g.*, [Bernasconi et al., 2022]), we assume that, if the receiver does *not* follow recommendations at some step $h \in \mathcal{H}$ by playing an action $\widehat{a}_h \neq a_h$, then the sender stops issuing future recommendations to the receiver.

For ease of presentation, we introduce the sender's value function $V_h^{\mathsf{S}, \phi} : \mathcal{T}_h \to \mathbb{R}$ to encode sender's expected rewards by using a history-dependent signaling scheme $\phi := \{\phi_\tau\}_{\tau \in \mathcal{T}}$ from step $h \in \mathcal{H}$ onwards, assuming the receiver always follows recommendations. Given a history $\tau = (s_1, a_1, \ldots, s_{h-1}, a_{h-1}, s_h) \in \mathcal{T}_h$ up to step $h$, such a value function is recursively defined as:

$$V_h^{\mathsf{S}, \phi}(\tau) = \sum_{\theta \in \Theta} \sum_{a \in \mathcal{A}} \mu_h(\theta|s_h)\phi_\tau(a|\theta) \left( r_h^{\mathsf{S}}(a, s_h, \theta) + \sum_{s' \in \mathcal{S}} p_h(s'|s_h, a, \theta) V_{h+1}^{\mathsf{S}, \phi}(\tau \oplus (a, s')) \right).$$

Similarly, we introduce the receiver's action-value function $V_h^{\mathsf{R}, \phi} : \mathcal{A} \times \mathcal{T}_h \to \mathbb{R}$ to encode the receiver's expected rewards by following sender's action recommendations from step $h \in \mathcal{H}$ onwards. Formally, given a history $\tau = (s_1, a_1, \ldots, s_{h-1}, a_{h-1}, s_h) \in \mathcal{T}_h$ up to step $h$, the receiver's expected reward by following the recommendation to play $a \in \mathcal{A}$ is recursively defined as follows:

$$V_h^{\mathsf{R}, \phi}(a, \tau) = \sum_{\theta \in \Theta} \mu_h(\theta|s_h)\phi_\tau(a|\theta) \left( r_h^{\mathsf{R}}(a, s_h, \theta) + \sum_{s' \in \mathcal{S}} p_h(s'|s_h, a, \theta) V_{h+1}^{\mathsf{R}, \phi}(\tau \oplus (a, s')) \right),$$

---

[3]In this paper, we let $\Delta(X)$ be the set of all the probability distributions over a finite set $X$, with $d(x)$ denoting the probability assigned to $x \in X$ by a distribution $d \in \Delta(X)$. Moreover, given a function $f : X \to \Delta(Y)$ with $X, Y$ any two finite sets, for every $x \in X$ we denote by $f(y|x)$ the probability that $f(x)$ assigns to $y \in Y$.

[4]In this work, we use *direct* signaling schemes which send signals in the form of action recommendations for the receiver. This is w.l.o.g. by well-known revelation principle arguments [Kamenica and Gentzkow, 2011].

[5]By a simple revelation-principle-style argument, we can focus w.l.o.g. on signaling schemes which depend on histories that do *not* include the sequence of private observations observed by the sender.

where $V_h^{R,\phi} : \mathcal{T}_h \to \mathbb{R}$ is such that $V_h^{R,\phi}(\tau) = \sum_{a \in \mathcal{A}} V_h^{R,\phi}(a, \tau)$ for every $h \in \mathcal{H}$ and $\tau \in \mathcal{T}_h$.

Finally, we introduce an additional receiver's value function, denoted by $\widehat{V}_h^R : \mathcal{S} \to \mathbb{R}$, to encode receiver's expected rewards from step $h \in \mathcal{H}$ onwards *after having deviated* from recommendations. Formally, for every state $s \in \mathcal{S}$, such a value function is recursively defined as follows:

$$\widehat{V}_h^R(s) = \max_{a \in \mathcal{A}} \sum_{\theta \in \Theta} \mu_h(\theta|s) \left( r_h^R(a, s, \theta) + \sum_{s' \in \mathcal{S}} p_h(s'|s, a, \theta) \widehat{V}_{h+1}^R(s') \right),$$

where the maximum operator encodes the fact that the receiver plays so as to maximize future rewards without knowledge of realized sender's private observations after having deviated.[6]

By the revelation principle [Kamenica and Gentzkow, 2011], it is well known that in order to find an optimal signaling scheme it is possible to focus w.l.o.g. on (direct) history-dependent signaling schemes under which the receiver is always incentivized to follow sender's action recommendations. These are called *persuasive* signaling scheme, and they are formally defined as follows:

**Definition 1** ($\epsilon$-persuasiveness). *Let $\epsilon \geq 0$. A history-dependent signaling scheme $\phi := \{\phi_\tau\}_{\tau \in \mathcal{T}}$ is said to be $\epsilon$-persuasive if, for every step $h \in \mathcal{H}$, history $\tau = (s_1, a_1, \ldots, s_{h-1}, a_{h-1}, s_h) \in \mathcal{T}_h$ up to step $h$, and pair of actions $a, a' \in \mathcal{A}$, the following holds:*

$$V_h^{R,\phi}(a, \tau) \geq \sum_{\theta \in \Theta} \mu_h(\theta|s_h) \phi_\tau(a|\theta) \left( r_h^R(s_h, a', \theta) + \sum_{s' \in \mathcal{S}} p_h(s'|s_h, a', \theta) \widehat{V}_{h+1}^R(s') - \epsilon \right).$$

*Moreover, we say that the signaling scheme is* persuasive *if the conditions above hold for $\epsilon = 0$. We denote the set of all the persuasive signaling schemes by $\Phi$.*

Whenever the sender commits to an $\epsilon$-persuasive history-dependent signaling scheme, we assume that the receiver always follows the recommendation issued by the sender. Notably, the recommended action is guaranteed to provide the receiver expected rewards that are at worst $\epsilon$ less than those attained by the best possible action.

In conclusion, the goal is to find an *optimal* signaling scheme for the sender, which is one achieving a sender's expected reward (from step one) $V^{S,\phi}$ greater than or equal to $\mathsf{OPT}$, defined as follows:

$$\mathsf{OPT} := \max_{\phi \in \Phi} V^{S,\phi}, \quad \text{where we let } V^{S,\phi} := \sum_{s \in \mathcal{S}} \beta(s) V_1^{S,\phi}((s)).$$

## 3 History-dependent signaling schemes are necessary

Previous works studying Bayesian persuasion in MDPs [Wu et al., 2022, Gan et al., 2022a] focus on Markovian signaling schemes, in which the action recommendation at step $h$ only depends on the current state $s_h$ and private observation $\theta_h$. Indeed, considering this class of signaling schemes is sufficient to optimize the utility of a sender facing a myopic receiver. Here, we show that this is *not* the case when the receiver is farsighted. In particular, we show that non-stationary Markovian signaling scheme are suboptimal. To do so, we show that there exists an MDP (see Appendix A) in which the optimal persuasive signaling scheme is history-dependent.[7] Intuitively, an history-dependent signaling scheme can adjust action recommendations depending on the choices available to the receiver in previous steps. Thus, if the receiver had profitable opportunities in the past, the sender must provide a larger expected reward to the receiver in order to be persuasive. Otherwise, the sender can aggressively maximize their expected rewards irrespective of receiver's ones. Formally:

**Theorem 1.** *There exist instances in which a persuasive history-dependent signaling scheme guarantees sender's expected rewards strictly greater than that obtained by an optimal persuasive non-stationary Markovian signaling scheme.*

---

[6]Notice that, in all the steps reached after having deviated from recommendations, the receiver does *not* get any clue about the sender's private information, and, thus, their expected reward only depends on the current state (*not* on the history). In other words, after deviating the receiver is playing a new MDP in which, by taking expectations with respect to $\mu_h$, rewards and transition probabilities are defined as $\tilde{r}_h(s, a) := \sum_{\theta \in \Theta} \mu_h(\theta|s) r_h^R(s, a, \theta)$ and $\tilde{p}_h(s'|s, a) := \sum_{\theta \in \Theta} \mu_h(\theta|s) p_h(s'|s, a, \theta)$, respectively, for all $h \in \mathcal{H}$, $s \in \mathcal{S}$, $a \in \mathcal{A}$, and $s' \in \mathcal{S}$.

[7]We defer to Appendix A the complete description of the instance, together with calculations of the optimal signaling schemes (for each class) and their corresponding value functions.

Moreover, we show that optimal non-stationary Markovian signaling schemes are NP-hard to approximate in polynomial time, even when the persuasiveness requirement is relaxed. This further motivates the use of history-dependent signaling schemes when addressing Bayesian persuasion problems in finite-horizon MDPs with a farsighted receiver. Formally, we prove the following:

**Theorem 2.** *There exist two constants $\alpha < 1$ and $\epsilon > 0$ such that computing an $\epsilon$-persuasive non-stationary Markovian signaling that provides the sender with at least a fraction $\alpha$ of the optimal sender's expected reward* OPT *is* NP-hard.

## 4 A sufficient subclass of efficiently-representable signaling schemes

Working with history-dependent signaling schemes begets unavoidable computational issues. These are due to the fact that explicitly representing such signaling schemes requires a number of bits growing exponentially in the size of the problem instance, since the number of possible histories is exponential in the time horizon $H$. In this section, we show how to circumvent such an issue by introducing a convenient subclass of signaling schemes—called *promise-form* signaling schemes— which are efficiently representable while being as good as history-dependent ones. In particular, our main result in this section (Theorem 3) shows that there always exists a promise-form signaling scheme which results in sender's expected rewards equal to its optimal value OPT.

### 4.1 Promise-form signaling schemes

A promise-form signaling scheme is defined by a set $\sigma := \{(I_h, \varphi_h, g_h)\}_{h \in \mathcal{H}}$ of triplets, where:

- $I_h : \mathcal{S} \to 2^{[0,H]}$ is a function defining, for every state $s \in \mathcal{S}$, a finite set $I_h(s) \subseteq [0, H]$ of *promises* for step $h \in \mathcal{H}$. We add the additional requirement that $0 \in I_1(s)$ for all $s \in \mathcal{S}$, and, for ease of notation, we set $I_{H+1}(s) := \{0\}$ for every $s \in \mathcal{S}$ and $\mathcal{I} := \bigcup_{h \in \mathcal{H}} \bigcup_{s \in \mathcal{S}} I_h(s)$.

- $\varphi_h : \mathcal{S} \times \mathcal{I} \times \Theta \to \Delta(\mathcal{A})$ is an *action-recommendation strategy* to be employed at step $h \in \mathcal{H}$, where $\varphi_h(a|s, \iota, \theta)$ is the probability of recommending action $a \in \mathcal{A}$ in state $s \in \mathcal{S}$ when the promise is $\iota \in I_h(s)$ and the sender's private observation is $\theta \in \Theta$.

- $g_h : \mathcal{S} \times \mathcal{A} \times \mathcal{I} \times \mathcal{S} \to \mathcal{I}$ is a *promise function* for step $h \in \mathcal{H}$ such that, whenever $h \leq H$ and $\iota \in I_h(s)$, $g_h(s, a, \iota, s') \in I_{h+1}(s')$ represents the promise for step $h+1$ if the next state is $s' \in \mathcal{S}$ and, at the current step $h$, action $a \in \mathcal{A}$ is recommended in state $s \in \mathcal{S}$.[8]

Intuitively, the rationale behind promise-form signaling schemes is that, when reaching a state $s_h \in \mathcal{S}$ at step $h \in \mathcal{H}$, the sender "promises" a value $\iota \in I_h(s_h)$ to the receiver, representing a lower bound on future rewards obtained by following recommendations. Moreover, sender's action recommendations only depend on the current state $s_h \in \mathcal{S}$, the sender's private observation $\theta_h \in \Theta$, and the current promise $\iota \in I_h(s_h)$, through the distribution $\varphi_h(s_h, \iota_h, \theta_h)$. Notice that it is always possible to infer the current promise by looking at the history of past states and action recommendations, by "composing" the functions $g_{h'}$ for the steps $h' < h$. This crucially avoids having to specify an explicit dependency on the full history of past states and action recommendations.

Notice that a promise-form signaling scheme as defined above does *not* automatically guarantee that the sender *honestly* keeps their promises. Indeed, in order to ensure that this is the case, we need to enforce additional constraints on the components of the signaling scheme, as we show in Section 4.3.

Let us remark that representing promise-form signaling schemes requires a number of bits polynomial in the size of the problem instance and in $|\mathcal{I}|$, which is the cardinality of the set of promises. While $|\mathcal{I}|$ could be arbitrarily large in general, the algorithm that we will present in the following Section 5 guarantees that $|\mathcal{I}|$ has "small" size, by means of a clever choice of the functions $I_h$.

### 4.2 From promise-form to history-dependent signaling schemes

In the following, we show how the sender can implement promise-form signaling schemes, proving that they represent a subclass of history-dependent ones.

---

[8]Notice that, in order to completely specify a promise-form signaling scheme $\sigma := \{(I_h, \varphi_h, g_h)\}_{h \in \mathcal{H}}$, it is sufficient to specify the functions $\varphi_h$ and $g_h$, since the functions $I_h$ can always be inferred by looking at the images of the functions $g_h$. However, we included $I_h$ in the definition of promise-form signaling schemes since this will considerably ease notation when dealing with them in Sections 5 and 6.

The sender can implement a promise-form signaling scheme $\sigma := \{(I_h, \varphi_h, g_h)\}_{h \in \mathcal{H}}$ as follows. After committing to $\sigma$ (Line 1 of Algorithm 1), at each step $h \in \mathcal{H}$, in Line 6 of Algorithm 1 they select which action-recommendation strategy to use by reconstructing the current promise on the basis of the history $\tau_h$. Such a reconstruction is done by recursively "composing" functions $g_{h'}$ for the preceding steps $h' < h$, by means of the procedure in Algorithm 2 with $\sigma$ and $\tau_h$ as inputs. By letting $\iota \in I_h(s_h)$ be the continuation value obtained by running Algorithm 2, the action recommendation $a_h$ in Line 6 is then sampled from $\varphi_h(s_h, \iota, \theta_h)$. Algorithm 2 clearly runs in time polynomial in the instance size, and, thus, the sender can implement a promise-form signaling scheme efficiently.

In the rest of this section, given a promise-form signaling scheme $\sigma := \{(I_h, \varphi_h, g_h)\}_{h \in \mathcal{H}}$, we denote by $\phi^\sigma := \{\phi_\tau^\sigma\}_{\tau \in \mathcal{T}}$ the history-dependent signaling scheme *induced* by $\sigma$ thorough the implementation procedure described above. Formally, for every history $\tau = (s_1, a_1, \ldots, s_{h-1}, a_{h-1}, s_h) \in \mathcal{T}_h$ up to step $h \in \mathcal{H}$, the function $\phi_\tau^\sigma : \Theta \to \Delta(\mathcal{A})$ is defined so that $\phi_\tau^\sigma(\theta) = \varphi(s_h, \iota_\tau^\sigma, \theta)$ for every $\theta \in \Theta$, where $\iota_\tau^\sigma \in I_h(s_h)$ denotes the promise value correspond-

---
**Algorithm 2** From histories to promises

**Require:** $\sigma := \{(I_h, \varphi_h, g_h)\}_{h \in \mathcal{H}}$,
$\quad \tau = (s_1, a_1, \ldots, s_{h-1}, a_{h-1}, s_h) \in \mathcal{T}_h$
1: Initialize $\iota \leftarrow 0 \in I_1(s_1)$
2: **for** each step $h' = 1, \ldots, h-1$ **do**
3: $\quad \iota \leftarrow g_{h'}(s_{h'}, a_{h'}, \iota, s_{h'+1})$
4: **return** $\iota$

---

ing to history $\tau$, as computed by Algorithm 2 with $\sigma$ and $\tau$ as inputs. As it is easy to see, implementing the promise-form signaling scheme $\sigma$ as described above is equivalent to using $\phi^\sigma$ in Algorithm 1.

Next, we show that the value functions of the sender and the receiver associated with the induced history-dependent signaling scheme $\phi^\sigma$ can be efficiently computed by only accessing the components of the promise-form signaling scheme $\sigma$. In the following, given any $\sigma := \{(I_h, \varphi_h, g_h)\}_{h \in \mathcal{H}}$, for every step $h \in \mathcal{H}$ we introduce the functions $\mathcal{V}_h^{\mathsf{R},\sigma} : \mathcal{A} \times \mathcal{S} \times \mathcal{I} \to \mathbb{R}$ and $\mathcal{V}_h^{\mathsf{R},\sigma} : \mathcal{S} \times \mathcal{I} \to \mathbb{R}$, which are jointly recursively defined so that, for every $a \in \mathcal{A}$, $s \in \mathcal{S}$, and $\iota \in I_h(s)$, it holds:

$$\mathcal{V}_h^{\mathsf{R},\sigma}(a, s, \iota) = \sum_{\theta \in \Theta} \mu_h(\theta|s)\varphi_h(a|s, \iota, \theta) \left( r_h^{\mathsf{R}}(s, a, \theta) + \sum_{s' \in \mathcal{S}} p_h(s'|s, a, \theta)\mathcal{V}_{h+1}^{\mathsf{R},\sigma}(s', g_h(s, a, \iota, s')) \right)$$

and $\mathcal{V}_h^{\mathsf{R},\sigma}(s, \iota) = \sum_{a \in \mathcal{A}} \mathcal{V}_h^{\mathsf{R},\sigma}(a, s, \iota)$. Similarly, $\mathcal{V}_h^{\mathsf{S},\sigma} : \mathcal{S} \times \mathcal{I} \to \mathbb{R}$ is such that, for $s \in \mathcal{S}, \iota \in I_h(s)$:

$$\mathcal{V}_h^{\mathsf{S},\sigma}(s, \iota) = \sum_{a \in \mathcal{A}} \sum_{\theta \in \Theta} \mu_h(\theta|s)\varphi_h(a|s, \iota, \theta) \left( r_h^{\mathsf{S}}(s, a, \theta) + \sum_{s' \in \mathcal{S}} p_h(s'|s, a, \theta)\mathcal{V}_{h+1}^{\mathsf{S},\sigma}(s', g_h(s, a, \iota, s')) \right).$$

Then, we can prove the following lemma:

**Lemma 1.** *Given a promise-form signaling scheme* $\sigma := \{(I_h, \varphi_h, g_h)\}_{h \in \mathcal{H}}$, *for every* $h \in \mathcal{H}$ *and history* $\tau = (s_1, a_1, \ldots, s_{h-1}, a_{h-1}, s_h) \in \mathcal{T}_h$ *up to step* $h$, *the following holds:*
$$V_h^{\mathsf{R},\phi^\sigma}(a, \tau) = \mathcal{V}_h^{\mathsf{R},\sigma}(a, s_h, \iota_\tau^\sigma), \; V_h^{\mathsf{R},\phi^\sigma}(\tau) = \mathcal{V}_h^{\mathsf{R},\sigma}(s_h, \iota_\tau^\sigma), \text{ and } V_h^{\mathsf{S},\phi^\sigma}(\tau) = \mathcal{V}_h^{\mathsf{S},\sigma}(s_h, \iota_\tau^\sigma).$$

Intuitively, Lemma 1 establishes that the functions $\mathcal{V}_h^{\mathsf{R},\sigma}$, $\mathcal{V}_h^{\mathsf{R},\sigma}$, and $\mathcal{V}_h^{\mathsf{S},\sigma}$ "correctly" encode the value functions of the sender and the receiver for a promise-form signaling scheme $\sigma$, when it is implemented according to the procedure described at the beginning of the section.

Finally, given how a promise-form signaling scheme $\sigma$ is implemented, in the following we say that $\sigma$ is $\epsilon$-persuasive (for some $\epsilon \geq 0$) if the induced history-dependent signaling scheme $\phi^\sigma$ is $\epsilon$-persuasive according to Definition 1. However, using such a definition to check whether $\sigma$ is $\epsilon$-persuasive is clearly computationally inefficient, since it would require working with exponentially-many histories. In Section 4.3, we introduce an easy way to ensure that a promise-form signaling scheme "keeps its promises", and we show that this allows to encode $\epsilon$-persuasiveness constraints in an efficient way.

## 4.3 The power of honesty

Next, we introduce a particular class of promise-form signaling schemes which always guarantee that the sender *honestly* (approximately) assures promised rewards to the receiver. We call $\eta$-*honest* the promise-form signaling schemes with such a property, which are formally defined as follows:

**Definition 2** ($\eta$-honesty)**.** *Let* $\eta \geq 0$. *A promise-form signaling scheme* $\sigma := \{(I_h, \varphi_h, g_h)\}_{h \in \mathcal{H}}$ *is $\eta$-honest if, for every step* $h \in \mathcal{H}$, *state* $s \in \mathcal{S}$, *and promise* $\iota \in I_h(s)$, *the following holds:*

$$\sum_{a \in \mathcal{A}} \sum_{\theta \in \Theta} \mu_h(\theta|s)\varphi_h(a|s, \iota, \theta) \left( r_h^{\mathsf{R}}(s, a, \theta) + \sum_{s' \in \mathcal{S}} p_h(s'|s, a, \theta)g_h(s, a, \iota, s') \right) \geq \iota - \eta. \quad (1)$$

Then, we can prove the following result on $\eta$-honest promise-form signaling schemes:

**Lemma 2.** *Let* $\sigma := \{(I_h, \varphi_h, g_h)\}_{h \in \mathcal{H}}$ *be a promise-form signaling scheme. If $\sigma$ is $\eta$-honest, then, for every step $h \in \mathcal{H}$ and state $s \in \mathcal{S}$, it holds that $\mathcal{V}_h^{R,\sigma}(s, \iota) \geq \iota - \eta(H - h + 1)$ for all $\iota \in I_h(s)$.*

Since by Lemma 1 the function $\mathcal{V}_h^{R,\sigma}$ encodes the receiver's value function when $\sigma$ is implemented by the sender, Lemma 2 intuitively establishes that, at each step $h \in \mathcal{H}$ and state $s \in \mathcal{S}$, the signaling scheme actually "keeps the promise" of giving at least $\iota \in I_h(s)$ future rewards to the receiver, up to an error depending on $\eta$. Checking whether a promise-form signaling scheme $\sigma := \{(I_h, \varphi_h, g_h)\}_{h \in \mathcal{H}}$ is $\eta$-honest or not can be done in time polynomial in the instance size and $|\mathcal{I}|$.

Now, we are ready to prove the following crucial lemma:

**Lemma 3.** *Let* $\sigma := \{(I_h, \varphi_h, g_h)\}_{h \in \mathcal{H}}$ *be an $\eta$-honest promise-form signaling scheme such that, for every $h \in \mathcal{H}$, $s \in \mathcal{S}$, $\iota \in I_h(s)$, and $a, a' \in \mathcal{A}$, the following constraint is satisfied:*

$$
\sum_{\theta \in \Theta} \mu_h(\theta|s) \varphi_h(a|s, \iota, \theta) \left( r_h^R(s, a, \theta) + \sum_{s' \in \mathcal{S}} p_h(s'|s, a, \theta) g_h(s, a, \iota, s') \right) \geq
$$
$$
\sum_{\theta \in \Theta} \mu_h(\theta|s) \varphi_h(a|s, \iota, \theta) \left( r_h^R(s, a', \theta) + \sum_{s' \in \mathcal{S}} p_h(s'|s, a', \theta) \widehat{V}_{h+1}^R(s') \right). \tag{2}
$$

*Then, we can conclude that $\sigma$ is $(\eta H)$-persuasive.*

Intuitively, Lemma 3 states that for an $\eta$-honest promise-form signaling scheme, the constraint in Equation (2) is equivalent to the one in Definition 1. The crucial advantage of Equation (2) is that it allows to express persuasiveness conditions as "local" constraints which do *not* require recursion.

### 4.4 Promise-form signaling schemes are sufficient

Finally, by exploiting Lemma 3, we can prove the main result of this section: promise-form signaling schemes represent a sufficient subclass of history-dependent ones. Formally:

**Theorem 3.** *There is always a persuasive promise-form signaling scheme* $\sigma := \{(I_h, \varphi_h, g_h)\}_{h \in \mathcal{H}}$ *with sender's expected reward equal to* OPT*. More formally, it holds that $V^{S,\phi^\sigma} =$ OPT *for the history-dependent signaling scheme $\phi^\sigma$ induced by $\sigma$.*

## 5 Approximation scheme

Theorem 3 shows that, in order to find an optimal signaling scheme, one can focus on promise-form signaling schemes that have the nice property of being polynomially representable in the instance size and the cardinality $|\mathcal{I}|$ of the set of promises. In this section, we show how to compute an $\epsilon$-persuasive promise-form signaling scheme with sender's expected rewards at least OPT in polynomial time.

We design an algorithm working with sets $I_h(s)$ defined on a suitable grid, whose size can be properly controlled by a discretization step $\delta$. The algorithm solves a recursively-defined optimization problem for each $h \in \mathcal{H}$, $s \in \mathcal{S}$, and $\iota \in I_h(s)$, by starting from step $H$ and proceeding in bottom up fashion.

For every step $h \in \mathcal{H}$ and state $s \in \mathcal{S}$, the set $I_h(s)$ of promises is defined as a suitable subset of a grid $\mathcal{D}_\delta := \{k \delta \mid k \in \mathbb{N} \wedge k \leq \lfloor H/\delta \rfloor\}$, where $\delta > 0$ is a discretization step to be set depending on the desired relaxation $\epsilon$ of the persuasiveness constraints. This also allows us to control the representation size of promise-form signaling schemes, as well as the running time of the algorithm. In particular, it holds $|\mathcal{D}_\delta| = O(1/\delta)$ and, thus, $|\mathcal{I}| = O(1/\delta)$ since $\mathcal{I} \subseteq \mathcal{D}_\delta$ by definition. In the following, for ease of notation, we let $\lceil x \rceil_\delta := \min_{k \in \mathbb{N}: k\delta \geq x} k\delta$ be the smallest multiple of $\delta$ greater than $x$, while $\lfloor x \rfloor_\delta := \max_{k \in \mathbb{N}: k\delta \leq x} k\delta$ is the greatest multiple of $\delta$ smaller than $x$.

The algorithm keeps track of recursively-computed values in a set of tables, one for each step. The table at step $h \in \mathcal{H}$ is encoded by means of a function $M_h^\delta : \mathcal{S} \times \mathcal{D}_\delta \to \mathbb{R} \cup \{-\infty\}$. Intuitively, for every $s \in \mathcal{S}$ and $\iota \in \mathcal{D}_\delta$, the entry $M_h^\delta(s, \iota)$ is related to the expected rewards achieved by the sender when "promising" the receiver expected rewards "approximately equal" to $\iota$ in state $s$ at step $h$. We also admit the functions $M_h^\delta$ to take value $-\infty$, which semantically corresponds to the case in which it is impossible to guarantee the promise $\iota$ to the receiver in state $s$ at step $h$. The entry $M_h^\delta(s, \iota)$ of

the table $M_h^\delta$ at step $h$ is computed recursively by solving a problem $\mathcal{P}_{h,s,\iota}(M_{h+1}^\delta)$ that we define in the following, where $M_{h+1}^\delta$ is the (previously-computed) table at step $h+1$.

By letting $M : \mathcal{S} \times \mathcal{D}_\delta \to \mathbb{R} \cup \{-\infty\}$ be a function encoding a generic table over $\mathcal{S} \times \mathcal{D}_\delta$, for every step $h \in \mathcal{H}$, state $s \in \mathcal{S}$, and promise $\iota \in I_h(s)$, we define the value $\Pi_{h,s,\iota}(M)$ of the optimization problem $\mathcal{P}_{h,s,\iota}(M)$ as follows:

$$\Pi_{h,s,\iota}(M) := \max_{\substack{\kappa:\Theta\to\Delta(\mathcal{A}) \\ q:\mathcal{A}\times\mathcal{S}\to\mathcal{D}_\delta}} F_{h,s,M}(\kappa,q) \quad \text{s.t.} \quad (\kappa,q) \in \Psi_\iota^{h,s},$$

where problem variables are encoded by the functions $\kappa : \Theta \to \Delta(\mathcal{A})$ and $q : \mathcal{A} \times \mathcal{S} \to \mathcal{D}_\delta$, which represent an action-recommendation strategy and a promise function, respectively.[9] The objective function $F_{h,s,M}(\kappa,q)$ of the optimization problem is defined as:

$$F_{h,s,M}(\kappa,q) := \sum_{\theta\in\Theta}\sum_{a\in\mathcal{A}} \mu_h(\theta|s)\kappa(a|\theta)\left(r_h^{\mathsf{S}}(s,a,\theta) + \sum_{s'\in\mathcal{S}} p_h(s'|s,a,\theta)M(s',q(a,s'))\right),$$

which encodes the sender's expected reward when their values for the next step $h+1$ are those specified by the table $M$. We assume that $0 \cdot (-\infty) = -\infty$. Moreover, the set $\Psi_\iota^{h,s}$ is comprised of the functions $\kappa : \Theta \to \Delta(\mathcal{A})$ and $q : \mathcal{A} \times \mathcal{S} \to \mathcal{D}_\delta$ that satisfy the following constraints:

$$\sum_{a\in\mathcal{A}}\sum_{\theta\in\Theta} \mu_h(\theta|s)\kappa(a|\theta)\left(r_h^{\mathsf{R}}(s,a,\theta) + \sum_{s'\in\mathcal{S}} p_h(s'|s,a,\theta)q(a,s')\right) \geq \iota \tag{4a}$$

$$\sum_{\theta\in\Theta} \mu_h(\theta|s)\kappa(a|\theta)\left(r_h^{\mathsf{R}}(s,a,\theta) + \sum_{s'\in\mathcal{S}} p_h(s'|s,a,\theta)q(a,s')\right) \geq$$

$$\sum_{\theta\in\Theta} \mu_h(\theta|s)\kappa(a|\theta)\left(r_h^{\mathsf{R}}(s,a',\theta) + \sum_{s'\in\mathcal{S}} p_h(s'|s,a',\theta)\widehat{V}_{h+1}^{\mathsf{R}}(s')\right) \qquad \forall a,a' \in \mathcal{A}, \tag{4b}$$

where Equation (4a) and Equation (4b) play the role of the honesty and the persuasiveness constraints, respectively. Notice that relaxing the honesty constraint yields larger feasible sets. Formally, for any $\iota \geq \iota' \geq 0$ we have that the following holds: $\Psi_\iota^{h,s} \subseteq \Psi_{\iota'}^{h,s}$.

If $F_{h,s,M}(\kappa,q) = -\infty$ for all $(\kappa,q) \in \Psi_\iota^{h,s}$, we have that $\Pi_{h,s,\iota}(M) = -\infty$. This intuitively comes from the fact that, if $\Pi_{h,s,\iota}(M) = -\infty$, then the value $\iota$ promised to the receiver is not realizable.

The optimization problem $\mathcal{P}_{h,s,\iota}(M)$ could be easily cast as a mixed-integer quadratic program, which are too general to be solved efficiently. Thus, we need specifically-tailored procedures to find an approximate solution to it. This discussion is deferred to Section 6. In the following, we assume to have access to an oracle $\mathcal{O}_{h,s,\iota}(M)$ that provides a suitable approximate solution to $\mathcal{P}_{h,s,\iota}(M)$. In particular, $\mathcal{O}_{h,s,\iota}(M)$ must satisfy the requirements introduced by the following definition:

**Definition 3** (Approximate Oracle). *An algorithm $\mathcal{O}_{h,s,\iota}(M)$ is an approximate oracle for $\mathcal{P}_{h,s,\iota}(M)$ if it returns a tuple $(\kappa,q,v)$ such that $\Pi_{h,s,\iota}(M) \leq v \leq F_{h,s,M}(\kappa,q)$ and $(\kappa,q) \in \Psi_{\iota-\delta}^{h,s}$.*

Intuitively, we ask that an approximate oracle finds a solution $(\kappa,q)$ in a slightly larger set $\Psi_{\iota-\delta}^{h,s}$. The oracle also returns a value $v$ to be inserted into $M_h^\delta(s,\iota)$, where $v$ is possibly different from the value $F_{h,s,M}(\kappa,q)$ of the objective function. This is needed for technical reasons in order to recover some concavity properties of the functions defining the tables used by the algorithm, which may be lost due to approximations. A complete discussion on this last aspect can be found in Section 6.

Equipped with an approximate oracle as in Definition 3, we are ready to design our approximation scheme that computes an $\epsilon$-persuasive promise-form signaling scheme attaining sender's expected reward at least $\mathsf{OPT}$ (Algorithm 3). The algorithm iteratively builds each table $M_h^\delta$ by filling it with the values $v$ returned by the approximate oracle. Moreover, it sets action-recommendation strategies $\varphi_h$ and promise functions $g_h$ of the signaling scheme $\sigma := \{(I_h, \varphi_h, g_h)\}_{h\in\mathcal{H}}$ to be equal to the functions $\kappa$, $q$ returned by the approximate oracle.

---

[9]The optimization problem over functions $\kappa$ and $q$ can be rewritten as an equivalent program with tabular variables, since the functions $\kappa$ and $q$ map discrete sets to discrete sets (or a randomization over them).

In order to clarify the semantic of the tables $M_h^\delta$ built by Algorithm 3, we prove the following preliminary result, which states that the sender's expected rewards for the signaling scheme returned by Algorithm 3 constitute an upper bound on the entries of the tables $M_h^\delta$. Formally:

**Lemma 4.** *Let $\sigma := \{(I_h, \varphi_h, g_h)\}_{h \in \mathcal{H}}$ be returned by Algorithm 3 instantiated with any oracle $\mathcal{O}_{h,s,\iota}$ as in Definition 3. For every $h \in \mathcal{H}$, $s \in \mathcal{S}$, $\iota \in I_h(s)$, it holds that $\mathcal{V}_h^{\mathsf{S},\sigma}(s,\iota) \geq M_h^\delta(s,\iota)$.*

Moreover, the entries of the tables $M_h^\delta$ built by Algorithm 3 are upper bounds on the sender's expected rewards provided by an optimal promise-form signaling scheme. Formally:

**Lemma 5.** *Let $M_h^\delta$, $I_h$ (for $h \in \mathcal{H}$) be computed by Algorithm 3 instantiated with any oracle $\mathcal{O}_{h,s,\iota}$ as in Definition 3, and let $\sigma^\star = \{(I_h^\star, \varphi_h^\star, g_h^\star)\}_{h \in \mathcal{H}}$ be an optimal promise-form signaling scheme. For every $h \in \mathcal{H}$, $s \in \mathcal{S}$, and $\iota \in I_h^\star(s)$, it holds that $\lceil \iota \rceil_\delta \in I_h(s)$ and $M_h^\delta(s, \lceil \iota \rceil_\delta) \geq \mathcal{V}_h^{\mathsf{S},\sigma^\star}(s,\iota)$.*

---

**Algorithm 3** Approximation scheme

**Require:** $\delta \in (0,1)$
1: $M_{H+1}^\delta(s,0) \leftarrow 0$ for all $s \in \mathcal{S}$
2: $M_{H+1}^\delta(s,\iota) \leftarrow -\infty$ for $s \in \mathcal{S}, \iota \in \mathcal{D}_\delta \setminus \{0\}$
3: **for** $h = H, \ldots, 1$ **do**
4:     **for** $s \in \mathcal{S}$ **do**
5:         $I_h(s) = \{\varnothing\}$
6:         **for** $\iota \in \mathcal{D}_\delta$ **do**
7:             $(\kappa, q, v) \leftarrow \mathcal{O}_{h,s,\iota-\delta}(M_{h+1}^\delta)$
8:             $M_h^\delta(s,\iota) \leftarrow v$
9:             $\varphi_h(a|s,\iota,\theta) \leftarrow \kappa(a|\theta)$
10:            $g_h(s,a,\iota,s') \leftarrow q(a,s')$
11:            **if** $v > -\infty$ **then**
12:               $I_h(s) \leftarrow I_h(s) \cup \{\iota\}$
13: **return** $\sigma := \{(I_h, \varphi_h, g_h)\}_{h \in \mathcal{H}}$

---

By combining Lemma 5 and Lemma 4 we can promptly state and prove the main result of this section.

**Theorem 4.** *For any $\epsilon > 0$, given an approximate oracle as in Definition 3, Algorithm 3 instantiated with $\delta = \epsilon/2H$ runs in time polynomial in $1/\epsilon$ and the instance size, while it finds an $\epsilon$-persuasive promise-form signaling scheme that guarantees expected reward at least $\mathsf{OPT}$ to the sender.*

# 6 Building a polynomial-time approximate oracle

In the previous section, we described an approximation scheme (Algorithm 3) that works provided it has access to an oracle that approximately solves the problem $\mathcal{P}_{h,s,\iota}(M)$ while satisfying the requirements of Definition 3. In this section, we exploit the specific structure of the problem to design an algorithm implementing such an oracle.

First, we relax the problem $\mathcal{P}_{h,s,\iota}(M)$ into $\mathcal{R}_{h,s,\iota}(M)$, which optimizes the expected value of the sender's rewards over *randomizations* of promises $\iota \in \mathcal{D}_\delta$. Moreover, we define for all $s' \in \mathcal{S}$ the set $\mathcal{D}_\delta(s', M) = \{\iota \in \mathcal{D}_\delta : M(s', \iota) > -\infty\}$ as the set of *realizable promises*, which semantically means

---

**Algorithm 4** Approximate oracle

**Require:** $\delta \in (0,1)$, $h \in \mathcal{H}$, $s \in \mathcal{S}$, $\iota \in \mathcal{D}_\delta$,     $M : \mathcal{S} \times \mathcal{D}_\delta \to \mathbb{R}$
1: **if** $\mathcal{R}_{h,s,\iota}(M)$ is feasible **then**
2:     $(\kappa, \tilde{q}) \leftarrow$ Solution to $\mathcal{R}_{h,s,\iota}(M)$
3:     $v \leftarrow \widetilde{F}_{h,s,M}(\kappa, \tilde{q})$
4:     $q \leftarrow \lfloor \tilde{q}_\mathbb{E} \rfloor_\delta$
5: **else**
6:     $(\kappa, q)$ arbitrary
7:     $v \leftarrow -\infty$
8: **return** $(\kappa, q, v)$

---

that those promises can be realized, conditioned on what is stored in the table $M$.[10] Specifically, the variables of $\mathcal{R}_{h,s,\iota}(M)$ are encoded by a function $\kappa : \Theta \to \Delta(\mathcal{A})$ and a randomized function $\tilde{q} : \mathcal{S} \times \mathcal{A} \to \Delta(\mathcal{D}_\delta)$. The objective function is

$$\widetilde{F}_{h,s,M}(\kappa,\tilde{q}) := \sum_{\theta \in \Theta} \sum_{a \in \mathcal{A}} \mu_h(\theta|s)\kappa(a|\theta)\left(r_h^{\mathsf{S}}(s,a,\theta) + \sum_{s' \in \mathcal{S}} p_h(s'|s,a,\theta) \sum_{\iota' \in \mathcal{D}_\delta(s',M)} \tilde{q}(\iota'|a,s')M(s',\iota')\right),$$

which modifies the objective function $F_{h,s,M}(\kappa,q)$ of $\mathcal{P}_{h,s,\iota}(M)$ by introducing the expectation over the possible promises $\iota' \in \mathcal{D}_\delta$, which are randomized through $\tilde{q}(\iota'|a,s')$.

To simplify notation, for any set $\mathcal{X}$, table $M$, and randomized function $\lambda : \mathcal{X} \times \mathcal{S} \to \Delta(\mathcal{D}_\delta)$, we denote by $\lambda_\mathbb{E} : \mathcal{X} \times \mathcal{S} \to \mathrm{co}(\mathcal{D}_\delta)$ a quantity related to its average, where co denotes the convex hull of a set. Formally, $\lambda_\mathbb{E}(x, s') = \sum_{\iota' \in \mathcal{D}_\delta(s', M)} \iota' \cdot \lambda(\iota'|x)$ for all $x \in \mathcal{X}$. Moreover, we denote by $\lfloor \lambda_\mathbb{E} \rfloor_\delta : \mathcal{X} \to \mathcal{D}_\delta$ its discrete average such that $\lfloor \lambda_\mathbb{E} \rfloor_\delta(x, s') = \lfloor \lambda_\mathbb{E}(x, s') \rfloor_\delta$ for all $x \in \mathcal{X}$ and

---

[10]Note that set is always not empty when instantiating $M = M_h^\delta$ for some $h \in \mathcal{H}$, as it always contains 0 for all $s' \in \mathcal{S}$. This can be easily seen from Lemma 5 with $\iota = 0$.

$s' \in \mathcal{S}$. Notice that, for ease of notation, we drop the dependence of the operator $\mathbb{E}$ from $M$, as such a dependence is always clear from the context.

By exploiting the notation introduced above, the relaxed optimization problem $\mathcal{R}_{h,s,\iota}(M)$ reads as:

$$\Omega_{h,s,\iota}(M) := \max_{\substack{\kappa:\Theta\to\Delta(\mathcal{A}) \\ \tilde{q}:\mathcal{A}\times\mathcal{S}\to\Delta(D_\delta)}} \widetilde{F}_{h,s,M}(\kappa,\tilde{q}) \quad \text{s.t.} \quad (\kappa, \tilde{q}_{\mathbb{E}}) \in \Psi_\iota^{h,s},$$

where we relax the functions $\tilde{q}$ to be distributions over $\mathcal{D}_\delta$, rather than deterministic. Then, we can prove the following:

**Lemma 6.** *The problem $\mathcal{R}_{h,s,\iota}(M)$ can be solved in time polynomial in $1/\delta$ and the instance size.*

By relying on a solution $(\kappa, \tilde{q})$ to $\mathcal{R}_{h,s,\iota}(M)$, Algorithm 4 is an approximate oracle for $\mathcal{P}_{h,s,\iota}(M)$. In particular, when the relaxed problem is feasible, the algorithm returns $k$ and the function $q$ obtained by de-randomizing the randomized function $\tilde{q}$ with its discrete average. Formally:

**Theorem 5.** *For every $h \in \mathcal{H}$, $s \in \mathcal{S}$, and $\iota \in \mathcal{D}_\delta$, if Algorithm 4 is used for all $h' > h$ as approximate oracle $\mathcal{O}_{h',s,\iota}$ in Algorithm 3, then it implements an approximate oracle as in Definition 3.*

Notice that the value $v$ returned by Algorithm 4 is the optimal value of the relaxed problem $\mathcal{R}_{h,s,\iota}(M)$. It is easy to prove (see the proof of Theorem 5) that the tables built by Algorithm 3 through Algorithm 4 are encoded by concave functions. This is the key feature that allows us to go from a randomized solution to its discrete average without loss in sender's expected rewards, and it is the reason why we need to assume that Algorithm 4 is employed at every step $h$ in Theorem 5.

## 7 Conclusion

In this paper, we provided a crucial advancement in Bayesian persuasion for sequential decision making, mastering the setting in which a farsighted receiver interacts with an MDP, a problem that was previously thought to be intractable [Wu et al., 2022, Gan et al., 2022a,b].

First, we have shown that the class of Markovian signaling schemes, the standard choice in previous works and most MDP settings, do not constitute the "right" class of policies, since finding an optimal Markovian signaling scheme is NP-hard. Instead, we demonstrated that history-dependent signaling schemes allow to both circumvent the negative complexity result affecting Markovian signaling schemes and guarantee higher expected rewards to the sender in general. Specifically, we designed an algorithm to find an optimal $\epsilon$-persuasive history-dependent signaling scheme in time polynomial in $1/\epsilon$ and in the instance size. The crucial component of the algorithm is to restrict the optimization to a convenient subclass of history-dependent signaling schemes, which we call *promise-form* as they encode the history into a promise of future rewards to the receiver. We showed that promise-form signaling schemes are both efficient to represent and as powerful as the general class of history-dependent signaling schemes. An interesting insight of our analysis reveals that being *honest*, *i.e.*, keeping promises of future rewards, is the key to persuade farsighted receivers in MDPs.

As a future work, we will investigate the online learning variant of the problem studied in this paper, as done in [Zu et al., 2021, Wu et al., 2022, Bernasconi et al., 2022], in which the transition function is *not* known. Moreover, it would be interesting to study the problem faced by a sender that needs to persuade a stream of receivers having adversarially-selected types, as done in [Balcan et al., 2015, Castiglioni et al., 2020b, 2021b, Bernasconi et al., 2023].

## Acknowledgements

This paper is supported by FAIR (Future Artificial Intelligence Research) project, funded by the NextGenerationEU program within the PNRR-PE-AI scheme (M4C2, Investment 1.3, Line on Artificial Intelligence), and by the EU Horizon project ELIAS (European Lighthouse of AI for Sustainability, No. 101120237).

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

# A  History-dependent signaling schemes are necessary

In this section, we provide an illustrative MDP instance in which an history-dependent signaling scheme is required in order to optimally solve a Bayesian persuasion problem with farsighted receiver. This instance also proves Theorem 1. The instance is depicted in Figure 1. It is easy to check that the receiver can achieve expected rewards

$$\widehat{V}_1^{\mathsf{R}}(s_0) = 5, \quad \widehat{V}_2^{\mathsf{R}}(s_1) = 10, \quad \widehat{V}_2^{\mathsf{R}}(s_2) = 0, \quad \widehat{V}_3^{\mathsf{R}}(s_3) = 0, \quad \widehat{V}_3^{\mathsf{R}}(s_4) = 0,$$

without sender's recommendations. If the sender commits to a *non-stationary Markovian* signaling scheme $\phi^\star = \{\phi_h\}_{h \in \mathcal{H}}$, where $\phi_h : \mathcal{S} \times \Theta \to \Delta(\mathcal{A})$, they can achieve an expected reward of $V^{\mathsf{S},\phi^\star} = V_1^{\mathsf{S},\phi^\star}(s_0) = 25$ while being persuasive. The latter is obtained by recommending actions deterministically as follows:[11]

$$\phi_2^\star(s_2) = a_1, \quad \phi_3^\star(s_3, \theta_0) = a_0, \quad \phi_3^\star(s_3, \theta_1) = a_0.$$

Instead, an *history-dependent* signaling scheme $\phi^\dagger := \{\phi_\tau\}_{\tau \in \mathcal{T}}$ can provide the sender with an expected reward of $V^{\mathsf{S},\phi^\dagger} = V_1^{\mathsf{S},\phi^\dagger}(s_0) = 30$ while being persuasive. This can only be obtained by adapting the action recommendation strategy in state $s_3$ according to whether the receiver passed through state $s_1$ or state $s_2$, as follows:

$$\phi_{(s_0,s_1,s_3)}^\dagger(\theta_0) = a_0, \quad \phi_{(s_0,s_1,s_3)}^\dagger(\theta_1) = a_0, \quad \phi_{(s_0,s_2,s_3)}^\dagger(\theta_0) = a_0, \quad \phi_{(s_0,s_2,s_3)}^\dagger(\theta_1) = a_1,$$

which makes the recommendation $\phi_{(s_0,s_2)}^\dagger = a_0$ persuasive as well.

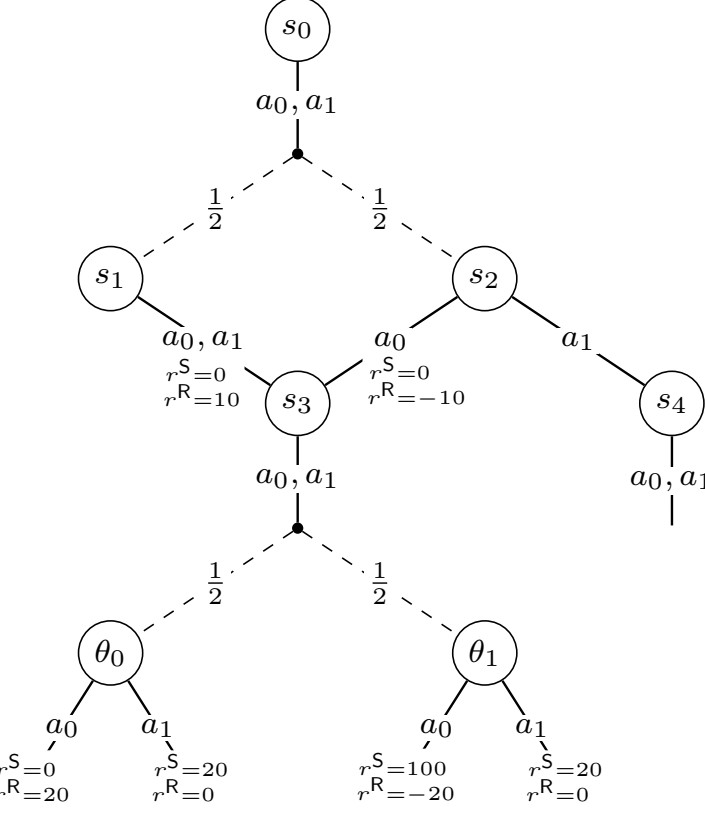

Figure 1: Visual representation of a Bayesian persuasion problem defined over an MDP with $\mathcal{S} = \{s_0, s_1, s_2, s_3\}, \mathcal{A} = \{a_0, a_1\}, \Theta = \{\theta_0, \theta_1\}, \mathcal{H} = [1, 2, 3]$, and $\beta(s_0) = 1$. The probabilities of the state transitions $p_h$ and the private observations $\mu_h$ are reported on the dashed edges. The reward functions $r_h^{\mathsf{S}}, r_h^{\mathsf{R}}$ for the sender and the receiver, respectively, are reported on the solid edges, when different from $0$. For the sake of clarity, the visualization omits some irrelevant information, such as private observations in $s_0, s_1, s_2, s_4$ and deterministic transitions.

---

[11]States where action selection is irrelevant are omitted for brevity.

The latter example proves the existence of an MDP instance where history-dependent signaling schemes are necessary. The reported instance does *not* require bizarre constructions, and we believe that exploiting history to optimize action recommendations is crucial in practical scenarios as well.

**Intuition of Figure 1.** In such a scenario, the MDP in Figure 1 can model a ride-sharing platform, where the drivers act as receivers and the platform is the sender. The platform can recommend routes and matching orders with the drivers, which then share with the platform a portion of the profit originating from the trips. In this example, $s_3$ can be thought as a location with high demand that is costly to reach, such as an airport terminal far from the city center. We can further think of the path $s_1 \rightarrow s_3$ as a driver's trip to the airport while serving an order, and $s_2 \rightarrow s_3$ as an empty trip instead. The platform can then adapt its order matching strategy according to whether the driver suffered empty costs or got a profit to come to the airport, such as guaranteeing a quick yet cheap match to the former, while waiting a more lucrative trip for the latter. This adaptive strategy can only be executed through an history-dependent signaling scheme. Instead, a Markovian signaling scheme cannot lure a driver into coming to the airport without serving an order, with diminishing profits for the platform.

# B  Proofs omitted from Section 3

**Theorem 2.** *There exist two constants $\alpha < 1$ and $\epsilon > 0$ such that computing an $\epsilon$-persuasive non-stationary Markovian signaling that provides the sender with at least a fraction $\alpha$ of the optimal sender's expected reward* OPT *is* NP-hard.

*Proof.* We reduce from a promise version of vertex cover in cubic graphs. In particular, given a cubic graph $(V, E)$, there exists a $\gamma \in (0, 2]$ such that it is NP-hard to decide whether there exists a vertex cover of size $k$ or all the vertex covers have size at least $(1 + \gamma)k$ [Alimonti and Kann, 2000].

We design an instance such that the set of states $\mathcal{S}$ includes $s_0$, $s_1$, $s_2$ and $s_3$. Moreover, it includes a state $s_e$ for each $e \in E$, and a state $s_v$ for each $v \in V$. The time horizon is set as $H = 3$. All the transition function and reward are independent from the time step $h \in [H]$. Hence, we will remove the subscript $h$ from the notation.

The rewards and transition in the different states are as follows:

- In state $s_0$ there is a single state of nature $\theta_0$ and two actions $a_{0,1}$ and $a_{0,2}$. The transition function is such that $p(s_3|s_0, a_{0,1}, \theta_0) = 1$ while $p(s_e|s_0, a_{0,2}, \theta_0) = \frac{1}{|E|}$ for each $e \in E$. The rewards of the receiver in state $s_0$ are $r^{\mathsf{R}}(a_{0,1}, s_0, \theta_0) = 1$ and $r^{\mathsf{R}}(a_{0,2}, s_0, \theta_0) = 0$. The rewards of the sender in state $s_0$ are $r^{\mathsf{S}}(a_{0,1}, s_0, \theta_0) = 0$ and $r^{\mathsf{S}}(a_{0,2}, s_0, \theta_0) = 1$.

- In state $s_1$, there is a single state of nature $\theta_1$ and a single actions $a_1$. The transition function is $p(s_2|s_1, a_1, \theta_1) = 1$, while the rewards of the sender and the receiver in $s_1$ are always 0.

- In state $s_2$, there is a single state of nature $\theta_2$ and a single actions $a_2$. The transition function is $p(s_v|s_2, a_2, \theta_2) = \frac{1}{|V|}$ for each $v \in V$. The sender's and receiver's rewards in $s_2$ are 0.

- In state $s_3$, there is a single state of nature $\theta_3$ and a single actions $a_3$. The transition function is such that $p(s_3|s_3, a_3, \theta_3) = 1$. The rewards of the sender and the receiver in $s_3$ are always 0.

- For each $e = (v, u) \in E$, in the state $s_e$, there is a single state of nature $\theta_e$, and two actions $a_{e,v}$ and $a_{e,u}$. The transition function is such that $p(s_v|s_e, a_{e,v}, \theta_e) = 1$ and $p(s_u|s_e, a_{e,u}, \theta_e) = 1$. The rewards of the sender and the receiver in state $s_e$ are 0.

- For each $v \in V$ in the state $s_v$, there are two states of nature $\theta_{v,1}$ and $\theta_{v,2}$. There are three actions $a_{v,1}$, $a_{v,2}$, and $a_{v,3}$. The transition function is such that $p(s_3|s_v, a, \theta) = 1$ for each $a$ and $\theta$. The rewards of the receiver in $s_v$ are $r^{\mathsf{R}}(s_v, a_{v,1}, \theta_{v,1}) = 1$, $r^{\mathsf{R}}(s_v, a_{v,2}, \theta_{v,2}) = 1$, $r^{\mathsf{R}}(s_v, a_{v,3}, \theta_{v,1}) = r^{\mathsf{R}}(s_v, a_{v,3}, \theta_{v,2}) = \frac{1}{2}$ and 0 otherwise. The rewards of the sender in $s_v$ is $\frac{1}{2}$ if the receiver plays $a_{v,3}$, *i.e.*, $r^{\mathsf{S}}(s_v, a_{v,3}, \theta_{v,1}) = r^{\mathsf{S}}(s_v, a_{v,3}, \theta_{v,2}) = \frac{1}{2}$, and 0 otherwise.

- All nature's states are drawn uniformly on the nature's outcome at each node.

- Finally, the initial distribution over the states is $\beta$ such that $\beta(s_0) = \beta(s_1) = \frac{1}{2}$.

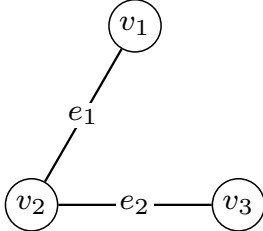

Figure 2: Illustrative instance of a cubic graph $(V, E)$.

In Figure 2 we have a simple instance of a cubic graph (undirected graph with degree bounded by 3), and in Figure 3 we have the instance the persuasion MDP built from the cubic graph in Figure 2.

We recall that a *non-stationary Markovian* signaling scheme is defined by a set of functions $\{\phi_h\}_{h \in \mathcal{H}}$, where $\phi_h : \mathcal{S} \times \Theta \to \Delta(\mathcal{A})$. However, we use both states $s_1$ and $s_2$ so that it takes two steps to arrive in states $s_v$ both when starting from $s_0$ and when starting from $s_1$. Hence, each state $s$ (excluding the sink state $s_3$) can be reached only with a specific time step $h$. This allows us to work only with stationary signaling scheme, *i.e.*, such that the signals do not depend on the time step. Hence, we can remove the subscript $h$ from the signaling scheme notation.

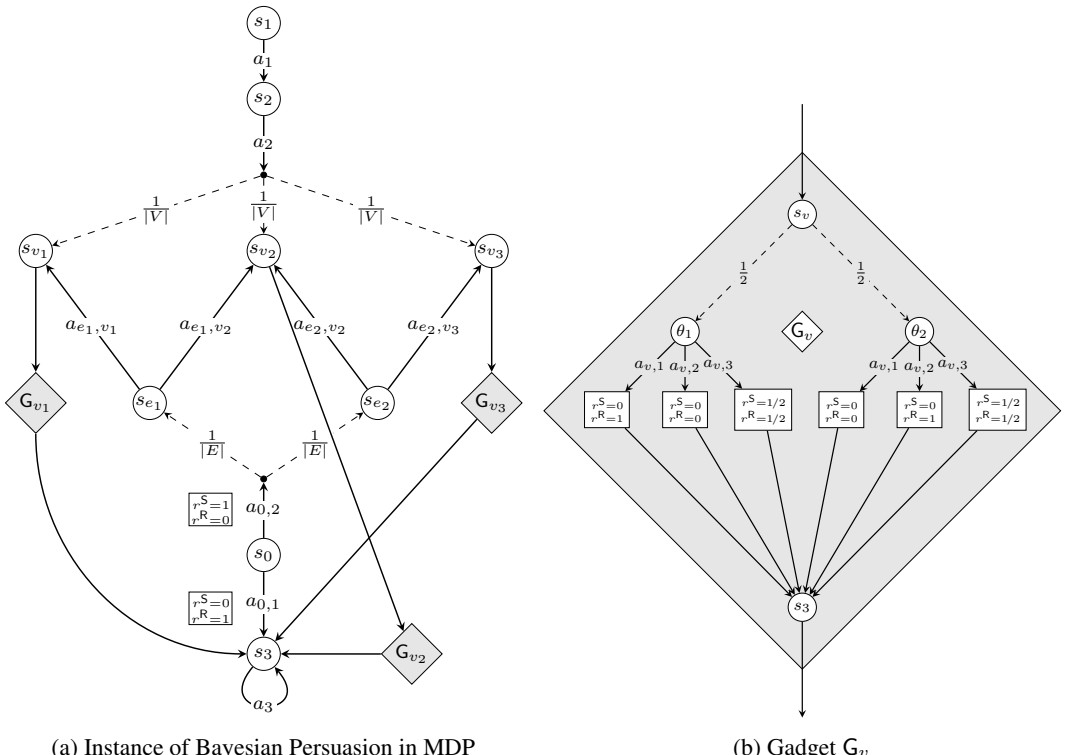

(a) Instance of Bayesian Persuasion in MDP    (b) Gadget $\mathsf{G}_v$

Figure 3: Instance of Bayesian persuasion in MDP constructed from the cubic graph $(V, E)$ of Figure 2. Sender's and receiver's rewards are reported only when different from 0. Solid lines represents actions (that are reported on the corresponding edges) while dashed lines represents stochastic outcomes (either of the transition model or of the prior). For each $v \in V$ the gadget $\mathsf{G}_v$ is reported Figure 3b.

We will show that if there exists a vertex cover of size $k$ then there exists a persuasive signaling scheme such that the sender's expected rewards is at least $\frac{3}{4} - \frac{k}{4|V|}$, while if all the vertex covers have size at least $(1 + \gamma)k$ then all the $\epsilon$-persuasive signaling schemes have sender's expected rewards strictly less than $\alpha \left( \frac{3}{4} - \frac{k}{4|V|} \right)$, where $\epsilon = \frac{1}{3} \cdot 10^{-4} \gamma^2$ and $\alpha = 1 - \frac{\gamma}{100}$. This will conclude the

proof as we could use an algorithm for our problem with approximation factor less or equal to $\alpha$ to distinguish between the two cases of the original problem.

**Completeness.** Suppose there exists a vertex cover $V^*$ of size $k$.

Consider the signaling scheme $\phi^*$ such that $\phi^*(a_{0,2}|s_0, \theta_0) = 1$, $\phi^*(a_1|s_1, \theta_1) = 1$, $\phi^*(a_2|s_2, \theta_2) = 1$, and $\phi^*(a_3|s_3, \theta_3) = 1$. Moreover, by the definition of vertex cover, for each $e \in E$ there exists a vertex $v_e \in V^*$ such that $v_e \in e = (v_e, u)$. We set $\phi^*(a_{e,v_e}|s_e, \theta_e) = 1$. Finally, for each $v \in V^*$, we set $\phi^*(a_{v,1}|s_v, \theta_{v,1}) = 1$, $\phi^*(a_{v,2}|s_v, \theta_{v,2}) = 1$, and $\phi^*(a_{v,3}|s_v, \theta_{v,1}) = \phi^*(a_{v,3}|s_v, \theta_{v,2}) = 1$ for all $v \notin V^*$.

First, we show that the signaling scheme is persuasive. Consider the state $s_0$. The receiver following the recommendations will always reach a state $v \in V^*$ and in this state will get utility 1. It is easy to see that playing action $a_{0,1}$ the utility is 1. Hence, in state $s_0$ the receiver will not deviate. In states $s_1$, $s_2$, and $s_3$, there is only one action available, hence the signaling scheme is trivially persuasive in these states. In a state $s_e$, $e \in E$, the receiver has no incentive to deviate. Indeed, following the recommendation they will transition to a state $s_v$, $v \in V^*$, and will get 1, while not following the recommendation they can get at most 1. In all the states $s_v$, for $v \in V^*$, the receiver gets 1, while deviating can get at most 1. Finally, in a state $s_v$, $v \notin V^*$, the receiver gets $\frac{1}{2}$, while deviating can get at most $\frac{1}{2}$. Hence, the signaling scheme is persuasive.

Moreover, the sender's expected rewards for playing according to $\phi^*$ is 1 if the initial state is $s_0$, as in this state the signaling scheme recommends $a_{0,2}$. On the other hand, if the initial state is $s_1$, then a state $s_v$ with $v \in V^*$ is reached with probability $k/|V|$ and a state $s_v$ with $v \notin V^*$ is reached with probability $1 - k/|V|$. Then, it is easy to see that the sender's expected rewards in reaching a state $s_v$ with $v \in V^*$ is 0 and $\frac{1}{2}$ otherwise. Thus the total sender's expected rewards of the signaling scheme $\phi^*$ is:

$$\frac{1}{2} + \frac{1}{4}\left(1 - \frac{k}{|V|}\right) = \frac{3}{4} - \frac{k}{4|V|}.$$

**Soundness.** Consider a signaling scheme $\phi$, and suppose that all the vertex covers have size at least $(1 + \gamma)k$. We show that the sender's expected rewards is strictly less than $\alpha\left(\frac{3}{4} - \frac{k}{4|V|}\right)$. First, notice that if the sender's expected rewards is at least $\alpha\left(\frac{3}{4} - \frac{k}{4|V|}\right)$ it has to recommend $a_{0,2}$ with probability at least $\ell := 3/10$, i.e., $\phi(a_{0,2}|s_0, \theta_0) \geq \frac{3}{10}$. Indeed, if this is not the case the sender's expected rewards is at most

$$\frac{1}{2}\frac{3}{10}\left(1 + \frac{1}{2}\right) + \frac{1}{2}\frac{1}{2} = \frac{19}{50} < \frac{\alpha}{2} \leq \alpha\left(\frac{3}{4} - \frac{k}{4|V|}\right),$$

as it gives at most $1 + \frac{1}{2}$ from $s_0$ when $a_{0,2}$ is played (which happens with probability less then $3/10$) and $\frac{1}{2}$ from $s_1$. Thus we can only consider the cases in which $\phi(a_{0,2}|s_0, \theta_0) \geq \frac{3}{10}$.

Then, suppose that for $n$ states $s_e$, $e \in E$, the receiver's expected rewards following the recommendations from there on is at most $1 - \delta$, where $\delta = \gamma/100$. Denote with $\widehat{E} \subseteq E$ be this set, where $|\widehat{E}| = n$, and with $\widetilde{E} := E \setminus \widehat{E}$. The $\epsilon$-persuasiveness constraint in state $s_0$ when the recommended action is $a_{0,2}$ implies:

$$\phi(a_{0,2}|s_0, \theta_0)\left[\frac{n}{|E|}(1 - \delta) + \left(1 - \frac{n}{|E|}\right)\right] \geq \phi(a_{0,2}|s_0, \theta_0)r^{\mathsf{S}}(a_{0,1}, s_0, \theta_0) - \epsilon,$$

as the left hand side is an upper bound on the receiver's expected rewards starting from $s_0$. The equation above implies that,

$$\frac{n}{|E|}(1 - \delta) + \left(1 - \frac{n}{|E|}\right) \geq 1 - \frac{\epsilon}{\ell},$$

and by rearranging it:

$$n \leq \frac{|E|\epsilon}{\ell\delta} = \frac{\frac{1}{3}|E|\gamma^2 10^{-4}}{3\gamma/1000} = \frac{\gamma|E|}{90}.$$

Let $\widetilde{V}$ be the set of states $s_v$, $v \in V$, such that the receiver's expected rewards from there on is at least $1 - \delta$. Notice that for each state $s_e$, with $e \in \widetilde{E}$, there exists a state $s_v$, with $v \in \widetilde{V}$, such that $v \in e$, *i.e.*, $v$ covers $e$. This implies that $\widetilde{V}$ covers $\widetilde{E}$.

By contradiction, we show that $|\widetilde{V}| \geq (1 + \gamma)k - n$. Suppose otherwise. Then, the set $\widetilde{V}$ is such that $|\widetilde{V}| < (1 + \gamma)k - n$ and covers at least $|\widetilde{E}|$ edges. Hence, there exists a super-set of $\widetilde{V}$ of size strictly smaller than $(1 + \gamma)k$ that covers all the edges, reaching a contradiction. This holds as adding a vertex $v$ for each $e = (v, u) \in \widehat{E}$ to $\widetilde{V}$, we obtain a vertex cover of $E$.

Then, we show that in all the states $s_v$, $v \in \widetilde{V}$ the sender's expected rewards from there on is at most $\delta$. It is easy to see that action $a_3$ is recommended with probability at most $2/\delta$ in each state $s_v$, $v \in \widetilde{V}$ and the sender collects $\frac{1}{2}$. If this is not the case, the receiver's expected rewards is strictly smaller than $2\delta\frac{1}{2} + (1 - 2\delta)1 = 1 - \delta$. This implies that the sender's expected rewards from a state $s_v$, $v \in \widetilde{V}$ is at most $\delta$. A similar argument shows that the sender's expected rewards from a state $s_e$, $e \in \widetilde{E}$, is at most $\delta$.

We conclude the proof by showing an upper bound on the sender's expected rewards. In the following, we will exploit that the graph is cubic and $|V| \geq k \geq \frac{|E|}{3}$. Moreover, we assume w.l.o.g. that there are no unconnected vertices, which implies that $|E| \geq \frac{|V|}{2}$ and thus that $k \geq \frac{|V|}{6}$. Hence, the sender's expected rewards when the initial state is $s_0$ is at most:

$$1 + \frac{1}{2}\frac{|\widehat{E}|}{|E|} + \delta\frac{|\widetilde{E}|}{|E|} = 1 + \frac{1}{2}\frac{n}{|E|} + \delta\left(1 - \frac{n}{|E|}\right) < 1 + \frac{\gamma}{180} + \delta.$$

Moreover, the sender's expected rewards when the initial state is $s_1$ is at most:

$$\delta\frac{|\widetilde{V}|}{|V|} + \frac{1}{2}\left(\frac{|V| - |\widetilde{V}|}{|V|}\right) \leq \delta\frac{(1+\gamma)k - n}{|V|} + \frac{1}{2}\left(1 - \frac{(1+\gamma)k - n}{|V|}\right)$$

$$= \frac{1}{2} - \left(\frac{1}{2} - \delta\right)\frac{(1+\gamma)k}{|V|} + \frac{1}{2}\frac{n}{|V|} - \frac{\delta n}{|V|}$$

$$< \frac{1}{2} - \left(\frac{1}{2} - \delta\right)\frac{(1+\gamma)k}{|V|} + \frac{1}{2}\frac{n}{|V|}$$

$$\leq \frac{1}{2} - \left(\frac{1}{2} - \delta\right)\frac{(1+\gamma)k}{|V|} + \frac{1}{2}\frac{\gamma|E|}{90|V|}$$

$$\leq \frac{1}{2} - \left(\frac{1}{2} - \delta\right)\frac{(1+\gamma)k}{|V|} + \frac{\gamma}{60},$$

where we used that $|\widetilde{V}| \geq (1 + \gamma)k - n$, $n \leq \gamma|E|/90$ and that $|E|/|V| \leq 3$.

Thus, since we start from $s_0$ or $s_1$ with probability $1/2$, the sender's expected rewards is at most:

$$\frac{1}{2}\left(1 + \frac{\gamma}{180} + \delta + \frac{1}{2} - \left(\frac{1}{2} - \delta\right)\frac{(1+\gamma)k}{|V|} + \frac{\gamma}{60}\right)$$

$$< \frac{3}{4} - \frac{k}{4|V|} + \frac{\gamma}{30} + \frac{\delta}{2}\left(1 + \frac{(1+\gamma)k}{|V|}\right) - \frac{\gamma k}{2|V|}$$

$$\leq \frac{3}{4} - \frac{k}{4|V|} + \frac{\gamma}{30} + \frac{\delta}{2}\left(1 + \frac{3k}{|V|}\right) - \frac{\gamma k}{2|V|}$$

$$\leq \frac{3}{4} - \frac{k}{4|V|} + \frac{\gamma}{30} + \frac{\delta}{2}\left(1 + 3\right) - \frac{\gamma k}{2|V|}$$

$$\leq \frac{3}{4} - \frac{k}{4|V|} + \frac{\gamma}{30} + \frac{\gamma}{50} - \frac{\gamma k}{2|V|}$$

$$\leq \frac{3}{4} - \frac{k}{4|V|} + \gamma\frac{8}{150} - \frac{\gamma k}{2|V|}$$

$$\leq \frac{3}{4} - \frac{k}{4|V|} + \gamma\frac{8}{150} - \frac{\gamma}{12}$$

$$\leq \frac{3}{4} - \frac{k}{4|V|} - \frac{3}{100}\gamma$$

$$\leq \frac{3}{4} - \frac{k}{4|V|} - \frac{3}{100}\gamma\left(\frac{3}{4} - \frac{k}{4|V|}\right)$$

$$\leq \left(1 - \frac{\gamma}{100}\right)\left(\frac{3}{4} - \frac{k}{4|V|}\right),$$

where we used that $k \geq |V|/6$ and $\delta := \gamma/100$.

This concludes the proof. $\qquad\square$

## C   Proofs omitted from Section 4

**Lemma 1.** *Given a promise-form signaling scheme $\sigma := \{(I_h, \varphi_h, g_h)\}_{h \in \mathcal{H}}$, for every $h \in \mathcal{H}$ and history $\tau = (s_1, a_1, \ldots, s_{h-1}, a_{h-1}, s_h) \in \mathcal{T}_h$ up to step $h$, the following holds:*

$$V_h^{R,\phi^\sigma}(a, \tau) = \mathcal{V}_h^{R,\sigma}(a, s_h, \iota_\tau^\sigma), \ V_h^{R,\phi^\sigma}(\tau) = \mathcal{V}_h^{R,\sigma}(s_h, \iota_\tau^\sigma), \text{ and } V_h^{S,\phi^\sigma}(\tau) = \mathcal{V}_h^{S,\sigma}(s_h, \iota_\tau^\sigma).$$

*Proof.* The statement can be easily proved by induction on the steps $\mathcal{H}$.

The base case of the induction is $h = H$. By definition of $V_H^{R,\phi^\sigma}(a, \tau)$, for every action $a \in \mathcal{A}$ and history $\tau = (s_1, a_1, \ldots, s_{H-1}, a_{H-1}, s_H) \in \mathcal{T}_H$, the following holds:

$$V_H^{R,\phi^\sigma}(a, \tau) = \sum_{\theta \in \Theta} \mu_H(\theta|s_H)\phi_\tau^\sigma(a|\theta)r_H^R(s_H, a, \theta).$$

Since by construction $\phi_\tau^\sigma(\theta) = \varphi_H(s_H, \iota_\tau^\sigma, \theta)$ for every $\theta \in \Theta$, we have that:

$$V_H^{R,\phi^\sigma}(a, \tau) = \sum_{\theta \in \Theta} \mu_H(\theta|s_H)\phi_\tau^\sigma(a|\theta)r_H^R(s_H, a, \theta)$$

$$= \sum_{\theta \in \Theta} \mu_H(\theta|s_H)\varphi_h(a|s_H, \iota_\tau^\sigma, \theta)r_H^R(s_H, a, \theta)$$

$$= \mathcal{V}_H^{R,\sigma}(a, s_H, \iota_\tau^\sigma).$$

Now, take $h < H$ and assume that the statement holds for $h + 1$. By definition of $V_h^{R,\phi^\sigma}(a, \tau)$, for every $a \in \mathcal{A}$ and $\tau = (s_1, a_1, \ldots, s_{h-1}, a_{h-1}, s_h) \in \mathcal{T}_h$, it holds:

$$V_h^{R,\phi^\sigma}(a, \tau) = \sum_{\theta \in \Theta} \mu_h(\theta|s_h)\phi_\tau^\sigma(a|\theta)\left(r_h^R(s_h, a, \theta) + \sum_{s' \in \mathcal{S}} p_h(s'|s_h, a, \theta)V_{h+1}^{R,\phi^\sigma}(\tau \oplus (a, s'))\right).$$

Moreover, for every $s' \in \mathcal{S}$, the relation $V_{h+1}^{R,\phi^\sigma}(\tau \oplus (a, s')) = \mathcal{V}_{h+1}^{R,\sigma}(s', \iota_{\tau \oplus (a,s')}^\sigma)$ holds by induction since $\tau \oplus (a, s')$ belongs to $\mathcal{T}_{h+1}$. Given how Algorithm 2 is designed, it holds:

$$\iota_{\tau \oplus (a,s')}^\sigma = g_h(s_h, a, \iota_\tau^\sigma, s'),$$

which, together with $\phi_\tau^\sigma(\theta) = \varphi_h(s_h, \iota_\tau^\sigma, \theta)$ for every $\theta \in \Theta$, gives $V_h^{R,\phi^\sigma}(a, \tau) = \mathcal{V}_h^{R,\sigma}(a, s_h, \iota_\tau^\sigma)$.

Similar inductive arguments show that $V_h^{S,\phi^\sigma}(\tau) = \mathcal{V}_h^{S,\sigma}(s_h, \iota_\tau^\sigma)$ for every step $h \in \mathcal{H}$ and history $\tau = (s_1, a_1, \ldots, s_{h-1}, a_{h-1}, s_h) \in \mathcal{T}_h$ up to $h$, concluding the proof. $\qquad\square$

**Lemma 2.** *Let $\sigma := \{(I_h, \varphi_h, g_h)\}_{h \in \mathcal{H}}$ be a promise-form signaling scheme. If $\sigma$ is $\eta$-honest, then, for every step $h \in \mathcal{H}$ and state $s \in \mathcal{S}$, it holds that $\mathcal{V}_h^{R,\sigma}(s, \iota) \geq \iota - \eta(H - h + 1)$ for all $\iota \in I_h(s)$.*

*Proof.* We prove the statement by induction.

The base case of the induction is $h = H$. Since $\sigma$ is $\eta$-honest, for every $s \in \mathcal{S}$ and $\iota \in I_H(s)$:

$$\mathcal{V}_H^{R,\sigma}(s, \iota) = \sum_{a \in \mathcal{A}} \sum_{\theta \in \Theta} \mu_H(\theta|s)\varphi_H(a|s, \iota, \theta)r_H^R(s, a, \theta) \geq \iota - \eta.$$

Now, take $h < H$ and assume that the statement holds for $h + 1$. For every $s \in \mathcal{S}$ and $\iota \in I_h(s)$:

$$\mathcal{V}_h^{\mathsf{R},\sigma}(s,\iota)$$

$$= \sum_{a \in \mathcal{A}} \sum_{\theta \in \Theta} \mu_h(\theta|s)\varphi_h(a|s,\iota,\theta) \left( r_h^{\mathsf{R}}(s,a,\theta) + \sum_{s' \in \mathcal{S}} p_h(s'|s,a,\theta)\mathcal{V}_{h+1}^{\mathsf{R},\sigma}(s',g_h(s,a,\iota,s')) \right)$$

$$\geq \sum_{a \in \mathcal{A}} \sum_{\theta \in \Theta} \mu_h(\theta|s)\varphi_h(a|s,\iota,\theta) \left( r_h^{\mathsf{R}}(s,a,\theta) + \sum_{s' \in \mathcal{S}} p_h(s'|s,a,\theta)(g_h(s,a,\iota,s') - \eta(H-h)) \right)$$

$$\geq \iota - \eta - \eta(H-h) \sum_{a \in \mathcal{A}} \sum_{\theta \in \Theta} \mu_h(\theta|s)\varphi_h(a|s,\iota,\theta) \sum_{s' \in \mathcal{S}} p_h(s'|s,a,\theta)$$

$$= \iota - \eta(H-h+1),$$

where the first inequality holds by induction, the second one by $\eta$-honesty, while the last equality holds since $\sum_{\theta \in \Theta} \mu_h(\theta|s)\varphi_h(a|s,\iota,\theta) \sum_{s' \in \mathcal{S}} p_h(s'|s,a,\theta) = 1$. This concludes the proof. $\square$

**Lemma 3.** *Let $\sigma := \{(I_h, \varphi_h, g_h)\}_{h \in \mathcal{H}}$ be an $\eta$-honest promise-form signaling scheme such that, for every $h \in \mathcal{H}$, $s \in \mathcal{S}$, $\iota \in I_h(s)$, and $a, a' \in \mathcal{A}$, the following constraint is satisfied:*

$$\sum_{\theta \in \Theta} \mu_h(\theta|s)\varphi_h(a|s,\iota,\theta) \left( r_h^{\mathsf{R}}(s,a,\theta) + \sum_{s' \in \mathcal{S}} p_h(s'|s,a,\theta)g_h(s,a,\iota,s') \right) \geq$$

$$\sum_{\theta \in \Theta} \mu_h(\theta|s)\varphi_h(a|s,\iota,\theta) \left( r_h^{\mathsf{R}}(s,a',\theta) + \sum_{s' \in \mathcal{S}} p_h(s'|s,a',\theta)\widehat{V}_{h+1}^{\mathsf{R}}(s') \right). \tag{2}$$

*Then, we can conclude that $\sigma$ is $(\eta H)$-persuasive.*

*Proof.* We recall that, for $\epsilon \geq 0$, the promise-form signaling scheme $\sigma$ is $\epsilon$-persuasive if the history-dependent signaling scheme $\phi^\sigma := \{\phi_\tau^\sigma\}_{\tau \in \mathcal{T}}$ induced by $\sigma$ is $\epsilon$-persuasive according to Definition 1.

As a consequence, in order to prove the statement, we have to show that, for every step $h \in \mathcal{H}$, history $\tau = (s_1, a_1, \ldots, s_{h-1}, a_{h-1}, s_h) \in \mathcal{T}_h$ up to step $h$, and pair of actions $a, a' \in \mathcal{A}$, it holds:

$$V_h^{\mathsf{R},\phi^\sigma}(a,\tau) \geq \sum_{\theta \in \Theta} \mu_h(\theta|s_h)\phi_\tau^\sigma(a|\theta) \left( r_h^{\mathsf{R}}(s_h,a',\theta) + \sum_{s' \in \mathcal{S}} p_h(s'|s_h,a',\theta)\widehat{V}_{h+1}^{\mathsf{R}}(s') - \epsilon \right).$$

By Lemma 1, we have that $V_h^{\mathsf{R},\phi^\sigma}(a,\tau) = \mathcal{V}_h^{\mathsf{R},\sigma}(a,s_h,\iota_\tau^\sigma)$. Moreover, the following holds:

$$\mathcal{V}_h^{\mathsf{R},\sigma}(a|s_h,\iota_\tau^\sigma)$$

$$= \sum_{\theta \in \Theta} \mu_h(\theta|s_h)\varphi_h(a|s_h,\iota_\tau^\sigma,\theta) \left( r_h^{\mathsf{R}}(s_h,a,\theta) + \sum_{s' \in \mathcal{S}} p_h(s'|s_h,a,\theta)\mathcal{V}_{h+1}^{\mathsf{R},\sigma}(s',g_h(s_h,a,\iota_\tau^\sigma,s')) \right)$$

$$\geq \sum_{\theta \in \Theta} \mu_h(\theta|s_h)\varphi_h(a|s_h,\iota_\tau^\sigma,\theta) \left( r_h^{\mathsf{R}}(s_h,a,\theta) + \sum_{s' \in \mathcal{S}} p_h(s'|s_h,a,\theta)\left(g_h(s_h,a,\iota_\tau^\sigma,s') - \eta(H-h)\right) \right)$$

$$\geq \sum_{\theta \in \Theta} \mu_h(\theta|s)\varphi_h(a|s_h,\iota_\tau^\sigma,\theta) \left( r_h^{\mathsf{R}}(s_h,a',\theta) + \sum_{s' \in \mathcal{S}} p_h(s'|s_h,a',\theta)\widehat{V}_{h+1}^{\mathsf{R}}(s') - \eta(H-h) \right)$$

where the first inequality holds by Lemma 2, while the second one holds thanks to Equation 2. Finally, the statement is readily proved by noticing that $\eta(H-h) \leq \eta H$. $\square$

**Theorem 3.** *There is always a persuasive promise-form signaling scheme $\sigma := \{(I_h, \varphi_h, g_h)\}_{h \in \mathcal{H}}$ with sender's expected reward equal to $\mathsf{OPT}$. More formally, it holds that $V^{\mathsf{S},\phi^\sigma} = \mathsf{OPT}$ for the history-dependent signaling scheme $\phi^\sigma$ induced by $\sigma$.*

*Proof.* The proof works by showing that, given any persuasive history-dependent signaling scheme, one can always build an $\eta$-honest and persuasive promise-form signaling scheme whose sender's expected reward is at least as good. This, together with Lemma 1 clearly proves the statement.

Given any persuasive history-dependent signaling scheme $\phi = \{\phi_\tau\}_{\tau \in \mathcal{T}}$, we build a promise-form signaling scheme $\sigma = \{(I_h, \varphi_h, g_h)\}_{h \in \mathcal{H}}$ as follows:

- $I_h(s) := \left\{ V_h^{\mathsf{R},\phi}(\tau) \mid \tau = (s_1, a_1, \ldots, s_{h-1}, a_{h-1}, s_h) \in \mathcal{T}_h \wedge s_h = s \right\}$ for all $h > 1$, $s \in \mathcal{S}$.

- $\varphi_h(s, \iota, \theta) = \phi_{\tau^\star_{h,s,\iota}}(\theta)$ for all $h \in \mathcal{H}$, $s \in \mathcal{S}$, $\iota \in I_h(s)$, and $\theta \in \Theta$, where:

$$\tau^\star_{h,s,\iota} \in \operatorname*{argmax}_{\substack{\tau = (s_1, a_1, \ldots, s_{h-1}, a_{h-1}, s_h) \in \mathcal{T}_h: \\ s_h = s \wedge V_h^{\mathsf{R},\phi}(\tau) = \iota}} \left\{ V_h^{\mathsf{S},\phi}(\tau) \right\},$$

  which is guaranteed to exist given how $I_h(s)$ is defined.

- $g_h(s, a, \iota, s') = V_{h+1}^{\mathsf{R},\phi}(\tau^\star_{h,s,\iota} \oplus (a, s'))$ for all $h \in \mathcal{H}$, $s \in \mathcal{S}$, $a \in \mathcal{A}$, $\iota \in I_h(s)$, and $s' \in \mathcal{S}$.

As a first step, we prove that, if $\phi$ is persuasive, then the promise-form signaling scheme $\sigma$ that we have just built is persuasive as well. This can be easily proved by exploiting Lemma 3.

First, we prove that $\sigma$ is an $\eta$-honest for $\eta = 0$. Formally, for every $h \in \mathcal{H}$, $s \in \mathcal{S}$, and $\iota \in I_h(s)$:

$$\iota = V_h^{\mathsf{R},\phi}(\tau^\star_{h,s,\iota}) \tag{6}$$

$$= \sum_{a \in \mathcal{A}} \sum_{\theta \in \Theta} \mu_h(\theta|s) \phi_{\tau^\star_{h,s,\iota}}(a|\theta) \left( r_h^{\mathsf{R}}(s, a, \theta) + \sum_{s' \in \mathcal{S}} p_h(s'|s, a, \theta) V_{h+1}^{\mathsf{R},\phi}(\tau^\star_{h,s,\iota} \oplus (a, s')) \right) \tag{7}$$

$$= \sum_{a \in \mathcal{A}} \sum_{\theta \in \Theta} \mu_h(\theta|s) \phi_{\tau^\star_{h,s,\iota}}(a|\theta) \left( r_h^{\mathsf{R}}(s, a, \theta) + \sum_{s' \in \mathcal{S}} p_h(s'|s, a, \theta) g_h(s, a, \iota, s') \right) \tag{8}$$

$$= \sum_{a \in \mathcal{A}} \sum_{\theta \in \Theta} \mu_h(\theta|s) \varphi_h(a|s, \iota, \theta) \left( r_h^{\mathsf{R}}(s, a, \theta) + \sum_{s' \in \mathcal{S}} p_h(s'|s, a, \theta) g_h(s, a, \iota, s') \right), \tag{9}$$

where Equation (6) holds by definition of $\tau^\star_{h,s,\iota}$, Equation (7) holds by definition of $V_h^{\mathsf{R},\phi}(\tau^\star_{h,s,\iota})$, Equation (8) holds by definition of $g_h(s, a, \iota, s')$, while Equation (9) holds by definition of $\varphi_h(a|s, \iota, \theta)$. This proves that $\sigma$ is an $\eta$-honest promise-form signaling scheme, for $\eta = 0$.

In order to apply Lemma 3, we also need to to prove that $\sigma$ satisfies the conditions in Equation (2). By applying definitions, it is easy to check that, for every $h \in \mathcal{H}$, $s \in \mathcal{S}$, $\iota \in I_h(s)$, and $a \in \mathcal{A}$:

$$V_h^{\mathsf{R},\phi}(a, \tau^\star_{h,s,\iota}) = \sum_{\theta \in \Theta} \mu_h(\theta|s) \phi_{\tau^\star_{h,s,\iota}}(a|\theta) \left( r_h^{\mathsf{R}}(s, a, \theta) + \sum_{s' \in \mathcal{S}} p_h(s'|s, a, \theta) V_{h+1}^{\mathsf{R},\phi}(\tau^\star_{h,s,\iota} \oplus (a, s')) \right)$$

$$= \sum_{\theta \in \Theta} \mu_h(\theta|s) \phi_{\tau^\star_{h,s,\iota}}(a|\theta) \left( r_h^{\mathsf{R}}(s, a, \theta) + \sum_{s' \in \mathcal{S}} p_h(s'|s, a, \theta) g_h(s, a, \iota, s') \right)$$

$$= \sum_{\theta \in \Theta} \mu_h(\theta|s) \varphi_h(a|s, \iota, \theta) \left( r_h^{\mathsf{R}}(s, a, \theta) + \sum_{s' \in \mathcal{S}} p_h(s'|s, a, \theta) g_h(s, a, \iota, s') \right). \tag{10}$$

Moreover, since $\phi$ is persuasive we have that, for every $h \in \mathcal{H}$, $s \in \mathcal{S}$, $\iota \in I_h(s)$, and $a, a' \in \mathcal{A}$:

$$V_h^{\mathsf{R},\phi}(a, \tau^\star_{h,s,\iota}) = \sum_{\theta \in \Theta} \mu_h(\theta|s) \phi_{\tau^\star_{h,s,\iota}}(a|\theta) \left( r_h^{\mathsf{R}}(s, a, \theta) + \sum_{s' \in \mathcal{S}} p_h(s'|s, a, \theta) V_{h+1}^{\mathsf{R},\phi}(\tau^\star_{h,s,\iota} \oplus (a, s')) \right)$$

$$\geq \sum_{\theta \in \Theta} \mu_h(\theta|s) \phi_{\tau^\star_{h,s,\iota}}(a|\theta) \left( r_h^{\mathsf{R}}(s, a', \theta) + \sum_{s' \in \mathcal{S}} p_h(s'|s, a', \theta) \widehat{V}_{h+1}^{\mathsf{R},\phi}(s') \right)$$

$$= \sum_{\theta \in \Theta} \mu_h(\theta|s) \varphi_h(a|s, \iota, \theta) \left( r_h^{\mathsf{R}}(s, a', \theta) + \sum_{s' \in \mathcal{S}} p_h(s'|s, a', \theta) \widehat{V}_{h+1}^{\mathsf{R},\phi}(s') \right), \tag{11}$$

where the last equality holds by definition of $\varphi_h(a|s, \iota, \theta)$. Then, by combining Equation (10) and Equation (11), we get that, for every $h \in \mathcal{H}$, $s \in \mathcal{S}$, $\iota \in I_h(s)$, and $a, a' \in \mathcal{A}$:

$$\sum_{\theta \in \Theta} \mu_h(\theta|s)\varphi_h(a|s, \iota, \theta) \left( r_h^{\mathsf{R}}(s, a, \theta) + \sum_{s' \in \mathcal{S}} p_h(s'|s, a, \theta)g_h(s, a, \iota, s') \right) \geq$$

$$\sum_{\theta \in \Theta} \mu_h(\theta|s)\varphi_h(a|s, \iota, \theta) \left( r_h^{\mathsf{R}}(s, a', \theta) + \sum_{s' \in \mathcal{S}} p_h(s'|s, a', \theta)\widehat{V}_{h+1}^{\mathsf{R},\phi}(s') \right),$$

which means that $\sigma$ satisfies Equation (2). Thus, by Lemma 3 we can conclude that $\sigma$ is persuasive.

In order to conclude the proof, it remains to show that $\sigma$ achieves a sender's expected reward at least as good as that obtained by $\phi$. Formally, we prove that $V_h^{\mathsf{S},\phi^\sigma}(\tau) \geq V_h^{\mathsf{S},\phi}(\tau)$ for every step $h \in \mathcal{H}$ and history $\tau \in \mathcal{T}_h$ up to $h$. We will prove such a result by induction.

The base case of the induction is $h = H$. For every $\tau = (s_1, a_1, \ldots, s_{H-1}, a_{H-1}, s_H) \in \mathcal{T}_H$:

$$V_H^{\mathsf{S},\phi}(\tau) \leq V_H^{\mathsf{S},\phi}(\tau_{H,s_H,\iota_\tau^\sigma}^\star)$$
$$= \sum_{a \in \mathcal{A}} \sum_{\theta \in \Theta} \mu_H(\theta|s_H)\phi_{\tau_{H,s_H,\iota_\tau^\sigma}^\star}(a|\theta)r_H^{\mathsf{S}}(s_H, a, \theta)$$
$$= \sum_{a \in \mathcal{A}} \sum_{\theta \in \Theta} \mu_H(\theta|s_H)\varphi_H(a|s_H, \iota_\tau^\sigma, \theta)r_H^{\mathsf{S}}(s_H, a, \theta)$$
$$= \mathcal{V}_H^{\mathsf{S},\sigma}(s_H, \iota_\tau^\sigma)$$
$$= V_H^{\mathsf{S},\phi^\sigma}(\tau),$$

where the last equality holds by Lemma 1.

Now, let us take $h < H$ and assume that the statement that we want to prove holds for $h + 1$. Then, for every $\tau = (s_1, a_1, \ldots, s_{h-1}, a_{h-1}, s_h) \in \mathcal{T}_h$, it holds:

$$V_h^{\mathsf{S},\phi}(\tau) \leq V_h^{\mathsf{S},\phi}(\tau_{h,s_h,\iota_\tau^\sigma}^\star) \tag{12}$$

$$= \sum_{a \in \mathcal{A}} \sum_{\theta \in \Theta} \mu_h(\theta|s_h)\phi_{\tau_{h,s_h,\iota_\tau^\sigma}^\star}(a|\theta) \left( r_h^{\mathsf{S}}(s, a, \theta) + \sum_{s' \in \mathcal{S}} p_h(s'|s, a, \theta)V_{h+1}^{\mathsf{S},\phi}(\tau_{h,s_h,\iota_\tau^\sigma}^\star \oplus (a, s')) \right)$$

$$= \sum_{a \in \mathcal{A}} \sum_{\theta \in \Theta} \mu_h(\theta|s_h)\varphi_h(a|s_h, \iota_\tau^\sigma, \theta) \left( r_h^{\mathsf{S}}(s_h, a, \theta) + \sum_{s' \in \mathcal{S}} p_h(s'|s_h, a, \theta)V_{h+1}^{\mathsf{S},\phi}(\tau_{h,s_h,\iota_\tau^\sigma}^\star \oplus (a, s')) \right)$$
$$\tag{13}$$

$$\leq \sum_{a \in \mathcal{A}} \sum_{\theta \in \Theta} \mu_h(\theta|s_h)\varphi_h(a|s_h, \iota_\tau^\sigma, \theta) \left( r_h^{\mathsf{S}}(s_h, a, \theta) + \sum_{s' \in \mathcal{S}} p_h(s'|s_h, a, \theta)V_{h+1}^{\mathsf{S},\phi}(\tau_{h+1,s',g_h(s_h,a,\iota_\tau^\sigma,s')}^\star) \right)$$
$$\tag{14}$$

$$\leq \sum_{a \in \mathcal{A}} \sum_{\theta \in \Theta} \mu_h(\theta|s_h)\varphi_h(a|s_h, \iota_\tau^\sigma, \theta) \left( r_h^{\mathsf{S}}(s_h, a, \theta) + \sum_{s' \in \mathcal{S}} p_h(s'|s_h, a, \theta)V_{h+1}^{\mathsf{S},\phi^\sigma}(\tau_{h+1,s',g_h(s_h,a,\iota_\tau^\sigma,s')}^\star) \right)$$
$$\tag{15}$$

$$\leq \sum_{a \in \mathcal{A}} \sum_{\theta \in \Theta} \mu_h(\theta|s_h)\varphi_h(a|s_h, \iota_\tau^\sigma, \theta) \left( r_h^{\mathsf{S}}(s_h, a, \theta) + \sum_{s' \in \mathcal{S}} p_h(s'|s_h, a, \theta)\mathcal{V}_{h+1}^{\mathsf{S},\sigma}(s', g_h(s_h, a, \iota_\tau^\sigma, s')) \right)$$
$$\tag{16}$$

$$= \mathcal{V}_h^{\mathsf{S},\sigma}(s_h, \iota_\tau^\sigma)$$
$$= V_h^{\mathsf{S},\phi^\sigma}(\tau). \tag{17}$$

where Equation (12) holds by definition of $\tau_{h,s_h,\iota_\tau^\sigma}^\star$ for the promise $\iota_\tau^\sigma \in I_h(s_h)$, Equation (13) holds by definition of $\varphi_h(s_h, \iota_\tau^\sigma, \theta)$, Equation (14) holds by definition of $\tau_{h+1,s',g_h(s_h,a,\iota_\tau^\sigma,s')}^\star$ together with the fact that $g_h(s_h, a, \iota_\tau^\sigma, s') = V_{h+1}^{\mathsf{R},\phi}(\tau_{h,s_h,\iota_\tau^\sigma}^\star \oplus (a, s'))$, Equation (15) holds by induction, while Lemma 1 proves Equation (16) and Equation (17).

The proof is completed by applying the definition of OPT. ◻

## D  Proofs omitted from Section 5

**Lemma 4.** *Let $\sigma := \{(I_h, \varphi_h, g_h)\}_{h \in \mathcal{H}}$ be returned by Algorithm 3 instantiated with any oracle $\mathcal{O}_{h,s,\iota}$ as in Definition 3. For every $h \in \mathcal{H}$, $s \in \mathcal{S}$, $\iota \in I_h(s)$, it holds that $\mathcal{V}_h^{\mathsf{S},\sigma}(s,\iota) \geq M_h^\delta(s,\iota)$.*

*Proof.* Let $\sigma = \{(I_h, \varphi_h, g_h)\}_{h \in \mathcal{H}}$ be the promise-form signaling scheme returned by Algorithm 3 instantiated with an approximate oracle $\mathcal{O}_{h,s,\iota}$ as in Definition 3, and let $(\kappa, q, v) \leftarrow \mathcal{O}_{h,s,\iota-\delta}(M_{h+1}^\delta)$. We prove the statement by induction. The base case for $h = H$ of the induction holds trivially. Then, for every $h < H$, $s \in \mathcal{S}$, and $\iota \in I_h(s)$, assuming the statement holds for $h + 1$ we have that:

$$\mathcal{V}_h^{\mathsf{S},\sigma}(s,\iota) = \sum_{a \in \mathcal{A}} \sum_{\theta \in \Theta} \mu_h(\theta|s) \varphi_h(a|s,\iota,\theta) \left( r_h^{\mathsf{S}}(s,a,\theta) + \sum_{s' \in \mathcal{S}} p_h(s'|s,a,\theta) \mathcal{V}_{h+1}^{\mathsf{S},\sigma}(s', g_h(s,a,\iota,s')) \right)$$

$$\geq \sum_{a \in \mathcal{A}} \sum_{\theta \in \Theta} \mu_h(\theta|s) \varphi_h(a|s,\iota,\theta) \left( r_h^{\mathsf{S}}(s,a,\theta) + \sum_{s' \in \mathcal{S}} p_h(s'|s,a,\theta) M_{h+1}^\delta(s', g_h(s,a,\iota,s')) \right)$$

$$= \sum_{a \in \mathcal{A}} \sum_{\theta \in \Theta} \mu_h(\theta|s) \kappa(a|\theta) \left( r_h^{\mathsf{S}}(s,a,\theta) + \sum_{s' \in \mathcal{S}} p_h(s'|s,a,\theta) M_{h+1}^\delta(s', q(a,s')) \right)$$

$$= F_{h,s,M_{h+1}^\delta}(\kappa, q)$$

$$\geq v = M_h^\delta(s,\iota)$$

where the first inequality holds by the the inductive assumption, while the second inequality holds thanks to Definition 3. ◻

**Lemma 5.** *Let $M_h^\delta$, $I_h$ (for $h \in \mathcal{H}$) be computed by Algorithm 3 instantiated with any oracle $\mathcal{O}_{h,s,\iota}$ as in Definition 3, and let $\sigma^\star = \{(I_h^\star, \varphi_h^\star, g_h^\star)\}_{h \in \mathcal{H}}$ be an optimal promise-form signaling scheme. For every $h \in \mathcal{H}$, $s \in \mathcal{S}$, and $\iota \in I_h^\star(s)$, it holds that $\lceil \iota \rceil_\delta \in I_h(s)$ and $M_h^\delta(s, \lceil \iota \rceil_\delta) \geq \mathcal{V}_h^{\mathsf{S},\sigma^\star}(s,\iota)$.*

*Proof.* We prove by induction that $M_h^\delta(s, \lceil \iota \rceil_\delta) \geq \mathcal{V}_h^{\sigma^\star}(s,\iota)$ for all $h \in \mathcal{H}$, $s \in \mathcal{S}$, and $\iota \in I_h^\star(s)$.

As a base for the induction, we consider $h = H$. Notice that $M_H(s, \lceil \iota \rceil_\delta)$ is an upper bound on the solution of $\mathcal{P}_{H,s,\lfloor \iota \rfloor_\delta}(M_{H+1}^\delta)$ (this holds by Definition 3). Moreover, $\hat{\kappa}(a|\theta) = \varphi_H^\star(a|s,\iota,\theta)$ and $\hat{q}(a,s') = 0$ is a feasible solution to $\mathcal{P}_{H,s,\lfloor \iota \rfloor_\delta}(M_{H+1}^\delta)$ and so $(\hat{\kappa}, \hat{q}) \in \Psi_{\lfloor \iota \rfloor_\delta}^{h,s} \subseteq \Psi_{\iota-\delta}^{h,s}$. This readily implies that $M_H^\delta(s, \lceil \iota \rceil_\delta) \geq \mathcal{V}_H^{\mathsf{S},\sigma^\star}(s,\iota)$, since:

$$M_H^\delta(s, \lceil \iota \rceil_\delta) = v$$
$$\geq \Pi_{h,s,\iota-\delta}(M_{H+1}^\delta)$$
$$= \max_{(\kappa,q) \in \Psi_{\iota-\delta}} F_{h,s,M_{H+1}^\delta}(\kappa, q)$$
$$\geq F_{h,s,M_{H+1}^\delta}(\hat{\kappa}, \hat{q}) = \mathcal{V}_H^{\sigma^\star}(s,\iota).$$

As for the inductive step, consider any $h < H$. First, we show that $\hat{\kappa}(a|\theta) = \varphi_h^\star(a|s,\iota,\theta)$ and $\hat{q}(a,s') = \lceil g_h^\star(s,a,\iota,s') \rceil_\delta$ is a feasible solution to $\mathcal{P}_{h,s,\lfloor \iota \rfloor_\delta}(M_{h+1}^\delta)$. Then, we show that it gets expected rewards larger than $\mathcal{V}_h^{\mathsf{S},\sigma^\star}(s,\iota)$.

First we prove feasibility of $(\hat{\kappa}, \hat{q})$. Consider first the constraint of Equation (4a).

$$\sum_{a \in \mathcal{A}} \sum_{\theta \in \Theta} \mu_h(\theta|s) \hat{\kappa}(a|\theta) \left( r_h^{\mathsf{R}}(s,a,\theta) + \sum_{s' \in \mathcal{S}} p_h(s'|s,a,\theta) \hat{q}(a,s') \right)$$

$$= \sum_{a \in \mathcal{A}} \sum_{\theta \in \Theta} \mu_h(\theta|s) \varphi_h^\star(a|s,\iota,\theta) \left( r_h^{\mathsf{R}}(s,a,\theta) + \sum_{s' \in \mathcal{S}} p_h(s'|s,a,\theta) \lceil g_h^\star(s,a,\iota,s') \rceil_\delta \right)$$

$$\geq \sum_{a \in \mathcal{A}} \sum_{\theta \in \Theta} \mu_h(\theta|s) \varphi_h^\star(a|s, \iota, \theta) \left( r_h^{\mathsf{R}}(s, a, \theta) + \sum_{s' \in \mathcal{S}} p_h(s'|s, a, \theta) g_h^\star(s, a, \iota, s') \right)$$

$$\geq \iota \geq \lfloor \iota \rfloor_\delta,$$

which proves that $(\hat{\kappa}, \hat{q})$ satisfies the constraint of Equation (4a).

Now, let us turn our attention to the constraint of Equation (4b):

$$\sum_{\theta \in \Theta} \mu_h(\theta|s) \hat{\kappa}(a|\theta) \left( r_h^{\mathsf{R}}(s, a, \theta) + \sum_{s' \in \mathcal{S}} p_h(s'|s, a, \theta) \hat{q}(a, s') \right)$$

$$= \sum_{\theta \in \Theta} \mu_h(\theta|s) \varphi_h^\star(a|s, \iota, \theta) \left( r_h^{\mathsf{R}}(s, a, \theta) + \sum_{s' \in \mathcal{S}} p_h(s'|s, a, \theta) \lceil g_h^\star(s, a, \iota, s') \rceil_\delta \right)$$

$$\geq \sum_{\theta \in \Theta} \mu_h(\theta|s) \varphi_h^\star(a|s, \iota, \theta) \left( r_h^{\mathsf{R}}(s, a, \theta) + \sum_{s' \in \mathcal{S}} p_h(s'|s, a, \theta) g_h^\star(s, a, \iota, s') \right)$$

$$\geq \sum_{\theta \in \Theta} \mu_h(\theta|s) \varphi_h^\star(a|s, \iota, \theta) \left( r_h^{\mathsf{R}}(s, a, \theta) + \sum_{s' \in \mathcal{S}} p_h(s'|s, a, \theta) \widehat{V}_{h+1}^{\mathsf{R}}(s') \right),$$

which proves $(\hat{\kappa}, \hat{q})$ satisfies the constraint of Equation (4b) and thus $\mathcal{P}_{h, s, \lfloor \iota \rfloor_\delta}(M_{h+1}^\delta)$ is feasible, i.e., $(\hat{\kappa}, \hat{q}) \in \Psi_{\lfloor \iota \rfloor_\delta} \subset \Psi_{\iota - \delta}$.

Now, we prove that it gets expected rewards larger than $\sigma^\star$.

$$M_h(s, \lceil \iota \rceil_\delta) = v$$
$$\geq \Pi_{H, s, \iota - \delta}(M_{h+1}^\delta)$$
$$= \max_{(\kappa, q) \in \Psi_{\iota - \delta}} F_{h, s, M_{h+1}^\delta}(\kappa, q)$$
$$\geq F_{h, s, M_{h+1}^\delta}(\hat{\kappa}, \hat{q})$$
$$= \sum_{\theta \in \Theta} \sum_{a \in \mathcal{A}} \mu_h(\theta|s) \hat{\kappa}(a|\theta) \left( r_h^{\mathsf{S}}(s, a, \theta) + \sum_{s' \in \mathcal{S}} p_h(s'|s, a, \theta) M_{h+1}^\delta(s', \hat{q}(a, s')) \right)$$
$$= \sum_{\theta \in \Theta} \sum_{a \in \mathcal{A}} \mu_h(\theta|s) \varphi_h^\star(a|\theta, \iota, \theta) \left( r_h^{\mathsf{S}}(s, a, \theta) + \sum_{s' \in \mathcal{S}} p_h(s'|s, a, \theta) M_{h+1}^\delta(s', \lceil g_h^\star(s, a, \iota, s') \rceil_\delta) \right)$$
$$\geq \sum_{\theta \in \Theta} \sum_{a \in \mathcal{A}} \mu_h(\theta|s) \varphi_h^\star(a|\theta, \iota, \theta) \left( r_h^{\mathsf{S}}(s, a, \theta) + \sum_{s' \in \mathcal{S}} p_h(s'|s, a, \theta) \mathcal{V}_{h+1}^{s, \sigma^\star}(s', g_h^\star(s, a, \iota, s')) \right)$$
$$= \mathcal{V}^{\mathsf{S}, \sigma^\star}(s', \iota)$$

where we used the induction assumption in the second inequality.

In conclusion, for every $\iota \in I_h^\star(s)$, it holds $\mathcal{V}_h^{\sigma^\star}(s, \iota) \geq 0$, and, thus, $M_h^\delta(s, \lceil \iota \rceil_\delta) \geq 0$, implying that $\mathcal{P}_{h, s, \iota - \delta}$ is feasible and thus $\iota \in I_h(s)$ $\square$

**Theorem 4.** *For any $\epsilon > 0$, given an approximate oracle as in Definition 3, Algorithm 3 instantiated with $\delta = \epsilon/2H$ runs in time polynomial in $1/\epsilon$ and the instance size, while it finds an $\epsilon$-persuasive promise-form signaling scheme that guarantees expected reward at least $\mathsf{OPT}$ to the sender.*

*Proof.* First of all we need to prove that the $\sigma$ returned by Algorithm 3 is indeed a promise-form signaling scheme. All the properties are trivially satisfied except for i) $g_h(s, a, \iota, s') \in I_{h+1}(s')$ for each $s \in \mathcal{S}, a \in \mathcal{A}, s' \in \mathcal{S}$, and $\iota \in I_h(s)$, and ii) $0 \in I_1(s)$ for each $s$.

We start proving the first property. Let $s \in \mathcal{S}, a \in \mathcal{A}, s' \in \mathcal{S}$, and $\iota \in I_h(s)$. Since $\iota \in I_h(s)$, by the definition of Algorithm 3 it holds that $v > -\infty$, where $(\kappa, q, v) \leftarrow \mathcal{O}_{h, s, \iota - \delta}(M_{h+1}^\delta)$ is the solution returned by the call to the oracle. It is easy to see that this implies that $M_{h+1}^\delta(s', q(a, s')) > -\infty$. We consider two cases.

- If $h = H$, then we have by construction that $M^\delta_{h+1}(s', q(a, s')) > -\infty$ if and only if $q(a, s') = 0$. Thus, $g_h(s, a, \iota, s') = q(a, s') = 0 \in I_{H+1}(s')$.

- If $h < H$, then we we $M^\delta_{h+1}(s', q(a, s')) > -\infty$ implying that $v' > -\infty$, where $(\kappa', q', v') \leftarrow \mathcal{O}_{h+1, s', q(a,s')-\delta}(M^\delta_{h+2})$ is the solution returned by the call to the oracle at step $h + 1$. By the definition of Algorithm 3, this implies that $q(a, s') \in I_{h+1}(s')$. Thus, $g_h(s, a, \iota, s') = q(a, s') \in I_{H+1}(s')$.

Now, we prove the second property. We only need to prove that for all $s \in \mathcal{S}$ we have $0 \in I_1(s)$ but this is easily proved by Lemma 5 instantiated for $h = 1$ and $\iota = 0$. Indeed we have that $M^\delta_1(s, 0) \geq \mathcal{V}^{\mathsf{S}, \sigma^\star}_h(s, 0)$, where we recall that $M^\delta_1(s, 0) = v$ and $(\kappa, q, v) \leftarrow \mathcal{O}_{h,s,-\delta}(M^\delta_2)$. By the definition of Algorithm 3 this implies that $0 \in I_1(s)$.

Then, we prove the persuasiveness. By Lemma 3 we only need to prove that $\sigma := \{(I_h, \varphi_h, g_h)\}_{h \in \mathcal{H}}$ satisfies the constraint of Equation 2 and that are $\eta$-honest for some $\eta$. This holds trivially for $\eta = 2\delta$ as for all $h \in \mathcal{H}, s \in \mathcal{S}$ and $\iota \in I_h(s)$ we have by construction that for all $h \in \mathcal{H}, s \in \mathcal{S}$ and $\iota \in I_h(s)$, and $(\kappa, q, v) \leftarrow \mathcal{O}_{h,s,\iota-\delta}(M^\delta_{h+1})$ we have

$$\varphi_h(a|s, \iota, \theta) = \kappa(a|\theta), \quad g_h(s, a, \iota, s') = q(a, s')$$

and by construction $(\kappa, q) \in \Psi^{h,s}_{\iota-2\delta}$. This implies that the constraints of Equation (4a) and Equation (4b) are verified and we can apply Lemma 3 which let us conclude that $\sigma$ is $(2\delta H)$-persuasive.

We now prove the optimality of the signaling scheme $\sigma$ returned by Algorithm 3. By combining Lemma 4 and Lemma 5 we have that for all $s \in \mathcal{S}, h \in \mathcal{H}$ and $\iota \in I^\star_h(s)$

$$\mathcal{V}^{\mathsf{S}, \sigma}_h(s, \lceil \iota \rceil_\delta) \geq \mathcal{V}^{\mathsf{S}, \sigma^\star}_h(s, \iota).$$

Then:

$$
\begin{aligned}
\mathsf{OPT} &\leq V^{\mathsf{S}, \phi^{\sigma^\star}} \\
&= \sum_{s \in \mathcal{S}} \beta(s) V^{\mathsf{S}, \phi^{\sigma^\star}}_1((s)) \\
&= \sum_{s \in \mathcal{S}} \beta(s) \mathcal{V}^{\mathsf{S}, \phi^{\sigma^\star}}_1(s, 0) \\
&\leq \sum_{s \in \mathcal{S}} \beta(s) \mathcal{V}^{\mathsf{S}, \sigma}_1(s, 0) \\
&= \sum_{s \in \mathcal{S}} \beta(s) V^{\mathsf{S}, \phi^\sigma}_1(s, 0) \\
&= V^{\mathsf{S}, \phi^\sigma},
\end{aligned}
$$

which proves our statement. $\qquad\square$

# E  Proofs omitted from Section 6

In this section we prove that the optimization problem $\mathcal{R}_{h,s,\iota}(M)$ admits a polynomial time algorithm that finds an exact solution. We rewrite here for clarity the problem $\mathcal{R}_{h,s,\iota}(M)$:

$$\max_{\substack{\kappa:\Theta\to\Delta(\mathcal{A}) \\ \tilde{q}:\mathcal{A}\times\mathcal{S}\to\Delta(\mathcal{D}_\delta)}} \sum_{\theta\in\Theta}\sum_{a\in\mathcal{A}} \mu_h(\theta|s)\kappa(a|\theta)\left( r^{\mathsf{S}}_h(s, a, \theta) + \sum_{s'\in\mathcal{S}} p_h(s'|s, a, \theta)\left( \sum_{\iota'\in\mathcal{D}_\delta(s',M)} \tilde{q}(\iota'|a, s')M(s', \iota') \right) \right)$$

(18a)

$$\text{s.t.} \sum_{a\in\mathcal{A}}\sum_{\theta\in\Theta} \mu_h(\theta|s)\kappa(a|\theta)\left( r^{\mathsf{R}}_h(s, a, \theta) + \sum_{s'\in\mathcal{S}} p_h(s'|s, a, \theta)\left( \sum_{\iota'\in\mathcal{D}_\delta(s',M)} \iota' \cdot \tilde{q}(\iota'|a, s') \right) \right) \geq \iota \quad (18b)$$

$$\sum_{\theta\in\Theta} \mu_h(\theta|s)\kappa(a|\theta)\left( r^{\mathsf{R}}_h(s, a, \theta) + \sum_{s'\in\mathcal{S}} p_h(s'|s, a, \theta)\left( \sum_{\iota'\in\mathcal{D}_\delta(s',M)} \iota' \cdot \tilde{q}(\iota'|a, s') \right) \right) \geq$$

$$\sum_{\theta\in\Theta}\mu_h(\theta|s)\kappa(a|\theta)\left(r_h^{\mathsf{R}}(s,a',\theta)+\sum_{s'\in\mathcal{S}}p_h(s'|s,a',\theta)\widehat{V}_{h+1}^{\mathsf{R}}(s')\right) \qquad \forall a,a'\in\mathcal{A}. \quad (18c)$$

Even if the optimization problem is defined over functions, clearly we can represent the functions $\kappa$ and $q$ with finite number of variables with linear constraints (to assure that the the function's outputs are distributions). In order to handle the quadratic terms of $\mathcal{R}_{h,s,\iota}(M)$ we define a "product" variable $z_{a,s',\iota}$ for every $a\in\mathcal{A}$, $s'\in\mathcal{S}$, and $\iota\in\mathcal{D}_\delta(s',M)$, which is used in place of

$$\sum_{\theta\in\Theta}\mu_h(\theta,s)\kappa(a|\theta)p_h(s'|s,a,\theta)\tilde{q}(\iota'|a,s')$$

in Program 18a and has to satisfy the constraint for all $a\in\mathcal{A}$ and $s'\in\mathcal{S}$:

$$\sum_{\iota'\in\mathcal{D}_\delta(s',M)}z_{a,s',\iota'}:=\sum_{\theta\in\Theta}\mu_h(\theta,s)\kappa(a|\theta)p_h(s'|s,a,\theta).$$

For the linear variable we can introduce the non-negative variables $\xi_{a,\theta}=\kappa(a|\theta)$ for each $\theta\in\Theta$ and $a\in\mathcal{A}$, that need to satisfy the simplex constraint for all $\theta\in\Theta$, *i.e.*, $\sum_{a\in\mathcal{A}}\xi_{a,\theta}=1$. By using these variables we can write the problem as an $\mathrm{LP}_{h,s,\iota}(M)$:

$$\max_{\substack{\xi_{a,\theta}\in\mathbb{R}_+,\\ z_{a,s',\iota'}\in\mathbb{R}_+}}\sum_{a\in\mathcal{A}}\sum_{\theta\in\Theta}\xi_{a,\theta}\mu_h(\theta|s)r_h^{\mathsf{S}}(s,a,\theta)+\sum_{s'\in\mathcal{S}}\sum_{a\in\mathcal{A}}\sum_{\iota'\in\mathcal{D}_\delta(s',M)}z_{a,s',\iota'}M(s',\iota') \quad (19a)$$

$$\text{s.t.} \quad \sum_{a\in\mathcal{A}}\sum_{\theta\in\Theta}\xi_{a,\theta}\mu_h(\theta|s)r_h^{\mathsf{R}}(s,a,\theta)+\sum_{s'\in\mathcal{S}}\sum_{a\in\mathcal{A}}\sum_{\iota'\in\mathcal{D}_\delta(s',M)}\iota'\cdot z_{a,s',\iota'}\geq\iota \quad (19b)$$

$$\sum_{\theta\in\Theta}\xi_{a,\theta}\mu_h(\theta|s)r_h^{\mathsf{R}}(s,a,\theta)+\sum_{s'\in\mathcal{S}}\sum_{\iota'\in\mathcal{D}_\delta(s',M)}\iota'\cdot z_{a,s',\iota'}\geq$$

$$\sum_{\theta\in\Theta}\xi_{a,\theta}\mu_h(\theta|s)\left(r_h^{\mathsf{R}}(s,a',\theta)+\sum_{s'\in\mathcal{S}}p_h(s'|s,a',\theta)\widehat{V}_{h+1}^{\mathsf{R}}(s')\right) \quad \forall a,a'\in\mathcal{A} \quad (19c)$$

$$\sum_{\iota'\in\mathcal{D}_\delta(s',M)}z_{a,s',\iota'}=\sum_{\theta\in\Theta}\mu_h(\theta,s)\xi_{a,\theta}p_h(s'|s,a,\theta) \qquad \forall s'\in\mathcal{S},\forall a\in\mathcal{A} \quad (19d)$$

$$\sum_{a\in\mathcal{A}}\xi_{a,\theta}=1 \qquad\qquad \forall\theta\in\Theta. \quad (19e)$$

Since $\mathrm{LP}_{h,s,\iota}(M)$ has polynomial number of variable and constraints one can find a solution in polynomial time.

The next two lemmas show that one can use $\mathrm{LP}_{h,s,\iota}(M)$. The first shows that any solution to $\mathcal{R}_{h,s,\iota}(M)$ can be used to find a solution to $\mathrm{LP}_{h,s,\iota}(M)$.

**Lemma 7.** *Let $(\kappa,q)\in\Psi_\iota^{h,s}$ be a feasible solution to $\mathcal{R}_{h,s,\iota}(M)$, then there exists a feasible solution to $\mathrm{LP}_{h,s,\iota}(M)$ with the same value.*

*Proof.* Let $(\kappa,\tilde{q})$ be a feasible solution to $\mathcal{R}_{h,s,\iota}(M)$. Then define:

$$z_{a,s',\iota'}:=\sum_{\theta\in\Theta}\mu_h(\theta,s)\kappa(a|\theta)p_h(s'|s,a,\theta)\tilde{q}(\iota'|a,s'),\forall a\in\mathcal{A},s'\in\mathcal{S},\iota'\in\mathcal{D}_\delta(s',M),$$

and

$$\xi_{a,\theta}:=\kappa(a|\theta),\forall a\in\mathcal{A},\theta\in\Theta.$$

Then one can show by direct calculation that all the constraints of $\mathrm{LP}_{h,s,\iota}(M)$ are satisfied and that the objective value of $\mathrm{LP}_{h,s,\iota}(M)$ is $\Omega_{h,s,\iota}(M)$ (which is the value of program $\mathcal{R}_{h,s,\iota}(M)$). $\qquad\square$

The next lemma show a result which is "complementary" to the one above.

**Lemma 8.** *Given a feasible solution of $\mathrm{LP}_{h,s,\iota}(M)$ one can find a solution to $\mathcal{R}_{h,s,\iota}(M)$ with at least the same value.*

*Proof.* Given a feasible solution $z$ and $\xi$ to $\text{LP}_{h,s,\iota}(M)$. Construct a solution $(\kappa, \tilde{q})$ to $\mathcal{R}_{h,s,\iota}(M)$ as follow for each $\iota' \in \mathcal{D}_\delta(s', M), a \in \mathcal{A}$ and $s \in \mathcal{S}$:

$$
\tilde{q}(\iota'|a, s') := \begin{cases} \frac{z_{a,s',\iota'}}{\sum_{\theta \in \Theta} \mu_h(\theta, s)\xi_{a,\theta} p_h(s'|s,a,\theta)} & \text{if } \sum_{\theta \in \Theta} \mu_h(\theta, s)\xi_{a,\theta} p_h(s'|s, a, \theta) > 0 \\ \mathbb{I}(\iota' = 0) & \text{otherwise} \end{cases}
$$

and for all $a \in \mathcal{A}$ and $\theta \in \Theta$:

$$
\kappa(a|\theta) = \xi_{a,\theta}.
$$

Notice that if for a specific $h \in \mathcal{H}, s \in \mathcal{S}, s' \in \mathcal{S}$ and $a \in \mathcal{A}$, the constraint of Equation (19d) impose that if $\sum_{\theta \in \Theta} \mu_h(\theta, s)\xi_{a,\theta} p_h(s'|s,a,\theta) = 0$ then $z_{a,s',\iota'} = 0$ for all $\iota'$. This means that in any case the following condition holds:

$$
z_{a,s',\iota'} = \sum_{\theta \in \Theta} \mu_h(\theta, s)\xi_{a,\theta} p_h(s'|s, a, \theta)\tilde{q}(\iota'|a, s'). \tag{20}
$$

We now fix any $h \in \mathcal{H}, s \in \mathcal{S}$ and $\iota \in \mathcal{D}_\delta$ show that the constraints of $\mathcal{R}_{h,s,\iota}(M)$ are satisfied, by using Equation (20). Let us start with the constraint of Equation (18b):

$$
\sum_{a \in \mathcal{A}} \sum_{\theta \in \Theta} \mu_h(\theta|s)\kappa(a|\theta) \left( r_h^{\mathsf{R}}(s, a, \theta) + \sum_{s' \in \mathcal{S}} p_h(s'|s, a, \theta) \left( \sum_{\iota' \in \mathcal{D}_\delta(s', M)} \iota' \cdot \tilde{q}(\iota'|a, s') \right) \right)
$$
$$
= \sum_{a \in \mathcal{A}} \sum_{\theta \in \Theta} \xi_{a,\theta} \mu_h(\theta|s) r_h^{\mathsf{R}}(s, a, \theta) + \sum_{a \in \mathcal{A}} \sum_{\theta \in \Theta} \sum_{\iota \in \mathcal{D}_\delta(s', M)} \sum_{s' \in \mathcal{S}} \iota' \xi_{a,\theta} \mu_h(\theta|s) p_h(s'|s, a, \theta) \tilde{q}(\iota'|a, s')
$$
$$
= \sum_{a \in \mathcal{A}} \sum_{\theta \in \Theta} \xi_{a,\theta} \mu_h(\theta|s) r_h^{\mathsf{R}}(s, a, \theta) + \sum_{a \in \mathcal{A}} \sum_{s \in \mathcal{S}} \sum_{\iota \in \mathcal{D}_\delta(s', M)} \iota' \cdot z_{a,s',\iota'}
$$
$$
\geq \iota,
$$

where the last inequality hold as $(z, \xi)$ is a feasible solution to $\text{LP}_{h,s,\iota}(M)$. This proves that the constraint of Equation (18b) is satisfied.

For Equation (18c) similarly:

$$
\sum_{\theta \in \Theta} \mu_h(\theta|s)\kappa(a|\theta) \left( r_h^{\mathsf{R}}(s, a, \theta) + \sum_{s' \in \mathcal{S}} p_h(s'|s, a, \theta) \left( \sum_{\iota' \in \mathcal{D}_\delta(s', M)} \iota' \cdot \tilde{q}(\iota'|a, s') \right) \right)
$$
$$
= \sum_{\theta \in \Theta} \xi_{a,\theta} \mu_h(\theta|s) r_h^{\mathsf{R}}(s, a, \theta) + \sum_{\theta \in \Theta} \sum_{\iota \in \mathcal{D}_\delta(s', M)} \sum_{s' \in \mathcal{S}} \iota' \xi_{a,\theta} \mu_h(\theta|s) p_h(s'|s, a, \theta) \tilde{q}(\iota'|a, s')
$$
$$
= \sum_{\theta \in \Theta} \xi_{a,\theta} \mu_h(\theta|s) r_h^{\mathsf{R}}(s, a, \theta) + \sum_{\theta \in \Theta} \sum_{\iota \in \mathcal{D}_\delta(s', M)} \sum_{s' \in \mathcal{S}} \iota' \cdot z_{a,s',\iota'}
$$
$$
\geq \sum_{\theta \in \Theta} \xi_{a,\theta} \mu_h(\theta|s) \left( r_h^{\mathsf{R}}(s, a', \theta) + \sum_{s' \in \mathcal{S}} p_h(s'|s, a', \theta) \widehat{V}_{h+1}^{\mathsf{R}}(s') \right),
$$

where in the last inequality we used that $(z, \xi)$ is a feasible solution to $\text{LP}_{h,s,\iota}(M)$. Thus the constraint of Equation (18c) is verified. This proves that $(\kappa, \tilde{q})$ is feasible for $\mathcal{R}_{h,s,\iota}(M)$. Then, by plugging Equation (20) into the objective of $\mathcal{R}_{h,s,\iota}(M)$ one directly prove that the values of the two problems is the same. $\qquad \square$

Now we are ready to prove the main result of this section:

**Lemma 6.** *The problem* $\mathcal{R}_{h,s,\iota}(M)$ *can be solved in time polynomial in* $1/\delta$ *and the instance size.*

*Proof.* Take a solution $(z, \xi)$ to $\text{LP}_{h,s,\iota}(M)$. This can be done in polynomial time as $\text{LP}_{h,s,\iota}(M)$ is a linear program with polynomial many variables and constraints. Then apply to the solution $(z, \xi)$ the trasformation used in Lemma 8 to obtain a feasible solution to $\mathcal{R}_{h,s,\iota}(M)$. This gives an optimal solution to $\mathcal{R}_{h,s,\iota}(M)$. To prove the last result, assume that there would exists a feasible solution $(\kappa', \tilde{q}')$ to $\mathcal{R}_{h,s,\iota}(M)$ such that $\tilde{F}_{h,s,M}(\kappa', \tilde{q}') > \tilde{F}_{h,s,M}(\kappa, \tilde{q})$, and apply the trasformation defined in Lemma 7. This would find a solution $(z', \xi')$ strictly better then $(z, \xi)$ which contradicts the optimality of $(z, \xi)$ for $\text{LP}_{h,s,\iota}(M)$. $\qquad \square$

**Theorem 5.** *For every $h \in \mathcal{H}$, $s \in \mathcal{S}$, and $\iota \in \mathcal{D}_\delta$, if Algorithm 4 is used for all $h' > h$ as approximate oracle $\mathcal{O}_{h',s,\iota}$ in Algorithm 3, then it implements an approximate oracle as in Definition 3.*

*Proof.* Fix any $\bar{h} \in \mathcal{H}$, $\bar{s} \in \mathcal{S}$ and $\bar{\iota} \in \mathcal{D}_\delta$. We show that for $h = \bar{h}+1$, $s \in \mathcal{S}$ and for any distribution $\gamma \in \Delta(\mathcal{D}_\delta)$, it holds that:

$$M_h^\delta \left( s, \lfloor \textstyle\sum_{\iota \in \mathcal{D}_\delta(s, M_h^\delta)} \iota\gamma(\iota) \rfloor_\delta \right) \geq \textstyle\sum_{\iota \in \mathcal{D}_\delta(s, M_h^\delta)} \gamma(\iota) M_h^\delta(s, \iota), \qquad (21)$$

where the table $M_h^\delta$ is the one built by Algorithm 3 with oracle of Algorithm 4.

Define $\iota_\mathbb{E} := \sum_{\iota \in \mathcal{D}_\delta(s, M_h^\delta)} \iota \cdot \gamma(\iota)$. Then for every $\iota, \iota' \in \mathcal{D}_\delta$ and $\iota' \leq \iota$ we have that

$$\Omega_{h,s,\iota'}(M_h^\delta) \geq \Omega_{h,s,\iota}(M_h^\delta),$$

as $\iota$ it only appears in the RHS of the constraint of the problem $\mathcal{R}_{h,s,\iota}(M_h^\delta)$.

By construction of Algorithm 3 we have that:

$$M_h^\delta(s, \iota) = \Omega_{h,s,\iota}(M_{h+1}^\delta),$$

and that by Lemma 7 and Lemma 8, $\Omega_{h,s,\iota}(M_{h+1}^\delta)$ is equal to the value of $\text{LP}_{h,s,\iota}(M)$ described by Equations (19), for each $\iota$. This means that the function $\iota \mapsto \Omega_{h,s,\iota}(M_{h+1}^\delta)$ is concave [Bertsekas, 1998, Theorem 5.1].

Thus for every distribution $\gamma \in \Delta(\mathcal{D}_\delta)$ we have:

$$\Omega_{h,s,(\sum_{\iota \in \mathcal{D}_\delta} \iota \cdot \gamma(\iota))}(M_{h+1}^\delta) \geq \sum_{\iota \in \mathcal{D}_\delta} \gamma(\iota) \cdot \Omega_{h,s,\iota}(M_{h+1}^\delta).$$

Observe that we cannot apply directly the above concavity property as, for every $s \in \mathcal{S}$, in Equation (21) we are only selecting the components of $\gamma$ such that $M_h^\delta > -\infty$.

To solve this problem we can, for any distribution $\gamma \in \Delta(\mathcal{D}_\delta)$, table $M$ and $s \in \mathcal{S}$. define the new distribution $\tilde{\gamma}$ on $\mathcal{D}_\delta(s, M)$ that puts all the mass on the $-\infty$ components of $M$ on 0. Formally $\tilde{\gamma}(\iota) = \gamma(\iota)$ for all $\iota \in \mathcal{D}_{\delta(s,M)}$, and $\tilde{\gamma}(0) = \gamma(0) + \sum_{\iota \in \mathcal{D}_\delta \setminus \mathcal{D}_\delta(s,M)} \gamma(\iota)$. Note that with this definition we have $\iota_\mathbb{E} = \sum_{\iota \in \mathcal{D}_\delta(s,M)} \iota \cdot \tilde{\gamma}(\iota)$ and $\sum_{\iota \in \mathcal{D}_\delta(s,M)} \gamma(\iota)\Omega_{h,s,\iota}(M) \leq \sum_{\iota \in \mathcal{D}_\delta(s,M)} \tilde{\gamma}(\iota)\Omega_{h,s,\iota}(M)$. Combining these inequalities we can conclude that:

$$\begin{aligned}
M_h^\delta(s, \lfloor \textstyle\sum_{\iota \in \mathcal{D}_\delta(s, M_h^\delta)} \iota\gamma(\iota) \rfloor_\delta) &= \Omega_{h,s,\lfloor \iota_\mathbb{E} \rfloor_\delta}(M_{h+1}^\delta) \\
&\geq \Omega_{h,s,\iota_\mathbb{E}}(M_{h+1}^\delta) \\
&\geq \sum_{\iota \in \mathcal{D}_\delta(s, M_h^\delta)} \tilde{\gamma}(\iota) \cdot \Omega_{h,s,\iota}(M_{h+1}^\delta) \\
&\geq \sum_{\iota \in \mathcal{D}_\delta(s, M_h^\delta)} \gamma(\iota) \cdot \Omega_{h,s,\iota}(M_{h+1}^\delta) \\
&= \sum_{\iota \in \mathcal{D}_\delta(s, M_h^\delta)} \gamma(\iota) \cdot M_h^\delta(s, \iota),
\end{aligned}$$

where the last inequality follows since it is easy to see that $\Omega_{h,s,0}(M_{h+1}^\delta) \geq 0$ and the last equality follows from Lemma 7. This proves Equation (21).

Now assume that $(\kappa, \tilde{q})$ is a the optimal solution to $\mathcal{R}_{\bar{h}, \bar{s}, \bar{\iota}}(M_{h+1}^\delta)$. Clearly if $(\kappa, \tilde{q}_\mathbb{E}) \in \Psi_{\bar{\iota}}^{\bar{h}, \bar{s}}$ then $(\kappa, q) \in \Psi_{\bar{\iota}-\delta}^{\bar{h}, \bar{s}}$, where $q = \lfloor \tilde{q}_\mathbb{E} \rfloor_\delta$.

Using the inequality of Equation (21) we can readily perform the following inequalities:

$$F_{\bar{h}, \bar{s}, M_{\bar{h}+1}^\delta}(\kappa, q)$$

$$= \sum_{\theta \in \Theta} \sum_{a \in \mathcal{A}} \mu_h(\theta|\bar{s})\kappa(a|\theta) \left( r_h^\mathsf{S}(\bar{s}, a, \theta) + \sum_{s' \in \mathcal{S}} p_h(s'|\bar{s}, a, \theta) M_{\bar{h}+1}^\delta(s', q(a, s')) \right)$$

$$\geq \sum_{\theta \in \Theta} \sum_{a \in \mathcal{A}} \mu_h(\theta|\bar{s}) \kappa(a|\theta) \left( r_h^{\mathsf{S}}(\bar{s}, a, \theta) + \sum_{s' \in \mathcal{S}} p_h(s'|\bar{s}, a, \theta) \sum_{\iota' \in \mathcal{D}_\delta(\bar{s}, M_{\bar{h}+1}^\delta)} \tilde{q}(\iota'|a, s') M_{\bar{h}+1}^\delta(s', \iota') \right)$$

$$= \tilde{F}_{\bar{h}, \bar{s}, M_{\bar{h}+1}^\delta}(\kappa, \tilde{q})$$

$$= v$$

$$= \Omega_{\bar{h}, \bar{s}, \bar{\iota}}(M_{\bar{h}+1}^\delta)$$

$$\geq \Pi_{\bar{h}, \bar{s}, \bar{\iota}}(M_{\bar{h}+1}^\delta)$$

and thus Algorithm 4 returns a tuple that satisfies the conditions required by Definition 3. □

