# OpenReview forum: "Persuading Farsighted Receivers in MDPs: the Power of Honesty"
_NeurIPS.cc/2023/Conference — NeurIPS 2023 poster_

### Official Review · Reviewer_XkbC · 2023-07-05

**Soundness:** 3 good
**Presentation:** 2 fair
**Contribution:** 2 fair
**Rating:** 5
**Confidence:** 2

**Summary:**

This paper studied a Bayesian persuasion problem where the sender and receiver act sequentially.

Below are two main changes in the new problem setting of this work:
(1) The authors assume that the sender stops providing recommendations to the receiver if the receiver does not follow the recommendation.
(2) Under this assumption, the authors also consider farsighted receivers as opposed to myopic receivers in previous works.

In the new setting, the authors showed that the Markovian signaling schemes are not optimal (additionally, finding the optimal Markovian signaling scheme in the previous problem settings in NP-hard), and they instead introduced a new class of promise-form signaling schemes for the new problem setting. The authors also show that the promise-form signaling schemes can be found in polynomial time while guaranteeing the schemes satisfy \epsilon-persuasive property.

**Strengths:**

1. Clearly listed all key theoretical findings under the assumption that the sender will
2. A novel class of promise-form signaling schemes is given with approximation algorithms that are relatively practical and can be completed in polynomial time

**Weaknesses:**

1. Overall a difficult-to-read paper (more difficult than the popular papers the authors cited) due to inadequate description and lack of concrete examples of the problem. Why sequential move in Bayesian persuasion is an important research topic is not highlighted. For a general audience not familiar with Bayesian persuasion, it is hard to tell why the new scenario is important.
2. Lack of justifications for the critical assumption that the sender will stop providing recommendations. Further justifications are needed to show that it is rational for the sender to prefer to stop recommending over other strategies commonly used like tit-for-tat, etc. If this is the typical case in real-world applications, the authors should also point out that to improve the soundness of this assumption. Since all findings in this paper are based on this assumption, I encourage the authors to put more emphasis on this.
3. Lack of discussion on the possible limitations


**Questions:**

1. Can you please provide application scenarios that fit well the new case
2. Can you please add a discussion on the potential limitations
3. In all problems that involve strategic manipulations, the potential fairness issues
4. Why are numerical experiments included in some of the related works, e.g., "Bayesian Persuasion in Sequential Decision-Making" but not here in this paper? What are some of the major differences that result in the decision of skipping the numerical experiment part?


**Limitations:**

Not that I can find due to my limited background in this topic.

---

> ### Author Rebuttal · Authors · 2023-08-09
>
> We want to thank the Reviewer for providing useful feedback despite being unfamiliar with the topic of the paper. We provide below detailed answers to their questions. Finally, we are sorry to hear that they find the paper to be a difficult read. We will make our best effort to make a final version of the paper accessible to a general audience despite its technical nature.
>
> 1. *“Can you please provide application scenarios that fit well the new case”*
>
> Due to space constraints, we deferred an explicative example in Appendix A (lines 436-446). We agree with the Reviewer that a concrete example would help understand the problem at hand and with the additional page provided in the final version of the paper we will include this example in the main body of the paper.
>
> 2. *“Can you please add a discussion on the potential limitations”*
>
> As the Reviewers Rxqa and q53d suggested, we will highlight the fact that our work assumes common knowledge of the environment, and we will discuss how this assumption could be lifted in future works.
>
> 3. *“In all problems that involve strategic manipulations, the potential fairness issues”*
>
> Research on the connection between information design and fairness is still understudied. While we agree with the Reviewer that it would be worth investigating, we feel that this topic falls outside the scope of our work.
>
> 4. *“Why are numerical experiments included in some of the related works, e.g., "Bayesian Persuasion in Sequential Decision-Making" but not here in this paper? What are some of the major differences that result in the decision of skipping the numerical experiment part?”*
>
> We think that the main contribution of this work is theoretical as we answer an open question in the literature. We will leave experimental evaluation of our algorithm to future research. Moreover the authors feel that an experimental evaluation of the planning problem would be less interesting compared to the one in [Bayesian Persuasion in Sequential Decision-Making], as we are not currently considering the learning problem.

---

### Official Review · Reviewer_Rxqa · 2023-07-07

**Soundness:** 4 excellent
**Presentation:** 4 excellent
**Contribution:** 4 excellent
**Rating:** 8
**Confidence:** 5

**Summary:**

The paper discusses a history-dependent signaling scheme in persuading a farsighted receiver. It first show that it is necessary for the sender to adopt a non-stationary and non-Markovian signaling scheme. Specifically, for every step and state reached at that step, this scheme defines a randomized mapping from the sender's private observations to action recommendations for the receiver, based on the whole history of states and receiver's actions observed up to that step. While such signaling policy could be intractable to describe, the paper provides a crucial simplification, promised-form signaling schemes, that allows the sender to only design finite size signaling scheme with optimal performance. Finally, the paper proposes a PTAS algorithm to determine the optimal promised-form signaling schemes.


**Strengths:**

1. The paper is very well written! The authors clearly explain the motivation, the model, the results, and the proofs with vivid intuitions, making the technical concepts very easy to follow.
2. The paper extends the previous work on signaling schemes in MDPs to a more general setting, where the receiver is farsighted. The paper made several important technical and conceptual contributions to this problem, including the necessity of non-stationary and non-Markovian signaling schemes, the simplification to the promised-form signaling schemes, and the PTAS algorithm to determine the optimal promised-form signaling schemes.

**Weaknesses:**

1. The method is related to the literature of dynamic stackelberg equilibrium. The authors should discuss the relationship between their methods.
2. I expect the authors to provide some real world applications of their model and methods, e.g., expand on the ride-sharing example in Appendix A.

**Questions:**

How does the author think of the learning problem under this farsighted setup? Is it also possible to design a no-regret learning algorithm for the sender to learn the optimal promised-form signaling scheme?

---

> ### Author Rebuttal · Authors · 2023-08-09
>
> We are grateful to hear that the Reviewer finds our paper to be making several important technical and conceptual contributions. We provide below a detailed answer to their question on the learning problem, while we will follow Reviewer’s suggestions to include a discussion of related literature on dynamic Stackelberg equilibrium and to expand the ride-sharing example in a final version of the paper.
>
> - *“How does the author think of the learning problem under this farsighted setup? Is it also possible to design a no-regret learning algorithm for the sender to learn the optimal promised-form signaling scheme?”*
>
> Our main focus with this work was to deal with the known model setting. Recent literature considers the learning problem in sequential BP settings, e.g. [Wu et al., 2022, Bernasconi et al., 2022, Gan et al., 2022a,b]. By extending our interactions model as in the above works we think that the techniques introduced there can also tackle the learning problem in our setting. In particular, we conjecture that an estimation phase that explores uniformly and then commits to an optimal signaling scheme of the estimated model would work. This should lead to an optimal $T^{2/3}$ bound on the regret and constraint violation. We will happily add a brief yet detailed discussion on this point.

---

> > ### Comment · Reviewer_Rxqa · 2023-08-18
> >
> > I appreciate the authors' detailed response. After reading the rebuttal and other reviews, I decide to maintain my initial score.

---

### Official Review · Reviewer_q53d · 2023-07-13

**Soundness:** 3 good
**Presentation:** 3 good
**Contribution:** 3 good
**Rating:** 7
**Confidence:** 4

**Summary:**

This paper considers a specific model of information design, where the receiver takes actions on a sequential decision process under a global, unknown natural state $\theta$. To make the model simple, the work assumes that the sequential decision process to be an MDP with a known model, plus the ability of the receiver to exactly optimize the cumulative reward once a (belief of) $\theta$ is given. In this case, the signaling scheme represents a (more general) mapping from the natural state and the trajectory up to the current step to a distribution of actions as information revelation. Further, when making decisions the receiver is not allowed to use the posterior distribution/belief of the natural state obtained from previous steps (which means only $\hat\theta_h$ is used).

Given the above assumptions in the model, the work provides a polynomial-time algorithm to obtain an $\epsilon$-persuasive signaling scheme. This disagrees with the NP-hard-like claims given in previous works. Such disagreement stems from the use of trajectory information and thereafter its simplified version of promise form. The algorithm is natural but quite creative.

**Strengths:**

1. The work provides new models of information design in sequential decision problems. The new model no longer possesses theoretical hardness.
2. The work proposes a new polynomial-time algorithm to find an $\epsilon$-persuasive signaling scheme.

**Weaknesses:**

Several limitations persist: 1) MDPs are assumed to be exactly solved. 2) MDP models are known 3) Receiver can't aggregate historical information of the natural state 4) No experiments.

**Questions:**

This conclusion indeed applies to the model-based scenario (meaning that the receiver knows the state set $S$, the observation set $\Theta$, the state transition distribution $p$, and the observation distribution $\mu$). However, does this conclusion also apply to the model-free case? This question might not be within the scope of the work, but the conclusion drawn is a bit too broad if such estimation of model is now involved. If this situation is not discussed, the authors should emphasize this important assumption in the abstract and introduction. Specifically, it should be clarified which information the receiver is assumed to know and base their decisions on.

This assumption represents a strong capability for the receiver. If it does not possess this knowledge, the sender's manipulation could be more powerful, and Markovian signaling schemes might be viable. For instance, if the receiver is unaware of the state and can only observe the sender's signals, and it needs to estimate the state and its transitions based on those signals, does the sender have the opportunity to confuse the receiver's judgments and achieve stronger persuasion? I find the claim "We consider the most general setting" in line 85 a bit too strong.

Additionally, can the revelation principle argument still be applied in a sequential setting? Does recommending only one action for each state achieve the goal of persuading a receiver in an MDP? I could not find any relevant discussion on this. Has the author considered "future-dependent" signaling schemes: recommending a set of future actions for each state instead of just one action? Or sending a signal $m$ to encode a set of future actions they wish to recommend? Moreover, in the aforementioned scenario, can the sender confuse multiple states by sending signal $m$? If the revelation principle is abandoned, is a history-dependent signaling scheme still necessary? If there is no discussion on the validity of the revelation principle, this assumption should be prominently emphasized.

**Limitations:**

See weakness

---

> ### Author Rebuttal · Authors · 2023-08-09
>
> We want to thank the Reviewer for their positive feedback. We report below detailed answers to their questions, which we hope will make them appreciate our paper even more.
>
> - *“This conclusion indeed applies to the model-based scenario (meaning that the receiver knows the state set $\mathcal{S}$, the observation set $\Theta$, the state transition distribution $p$, and the observation distribution $\mu$). However, does this conclusion also apply to the model-free case? This question might not be within the scope of the work, but the conclusion drawn is a bit too broad if such estimation of model is now involved. If this situation is not discussed, the authors should emphasize this important assumption in the abstract and introduction. Specifically, it should be clarified which information the receiver is assumed to know and base their decisions on.”*
>
> In this work we assume that everyone knows everything. We agree with the Reviewer that studying the learning problem in this setting is interesting and worth studying. However this imposes some difficulties as, when the model parameters are unknown, the receiver cannot even know if the signaling scheme employed by the sender is persuasive or not. See also response to Reviewer Rxqa for a related discussion.
>
> - *“This assumption represents a strong capability for the receiver. If it does not possess this knowledge, the sender's manipulation could be more powerful, and Markovian signaling schemes might be viable. For instance, if the receiver is unaware of the state and can only observe the sender's signals, and it needs to estimate the state and its transitions based on those signals, does the sender have the opportunity to confuse the receiver's judgments and achieve stronger persuasion? I find the claim "We consider the most general setting" in line 85 a bit too strong.”*
>
> If the receiver does not know the model, it is not clear which is the “right” definition of rationality. We can assume that the receiver “learns” during time but there are too many possible options (do they use confidence bounds? Do they use a regret minimizer? Do they follow the recommendations when they are uncertain or not?). On the other extreme, we can assume that the receiver never learns anything. This is the myopic setting studied in previous works.
>
> - *“Additionally, can the revelation principle argument still be applied in a sequential setting? Does recommending only one action for each state achieve the goal of persuading a receiver in an MDP? I could not find any relevant discussion on this. Has the author considered "future-dependent" signaling schemes: recommending a set of future actions for each state instead of just one action? Or sending a signal $m$ to encode a set of future actions they wish to recommend? Moreover, in the aforementioned scenario, can the sender confuse multiple states by sending signal $m$? If the revelation principle is abandoned, is a history-dependent signaling scheme still necessary? If there is no discussion on the validity of the revelation principle, this assumption should be prominently emphasized.”*
>
> We followed the recent extensive literature of information design (even in sequential settings, e.g. [Wu et al., 2022, Bernasconi et al., 2022, Gan et al., 2022a,b]) in which the signals are single action recommendations, and the revelation principle is more or less assumed to hold. A formal proof of the statement would require a large amount of notation and labor and deviate from the authors’ main focus with this work. Moreover, it would use standard techniques to derive a non-surprising result. However, we will happily underline this choice better in the final version of the paper.

---

> > ### Comment · Reviewer_q53d · 2023-08-21
> > **Response**
> >
> > I've read other reviews and the rebuttal. The evaluation remains the same with the rebuttal (as not much more information is provided in the rebuttal). I thank the authors for the response.

---

### Official Review · Reviewer_bcaS · 2023-07-19

**Soundness:** 3 good
**Presentation:** 3 good
**Contribution:** 2 fair
**Rating:** 6
**Confidence:** 5

**Summary:**

The paper considers a (finite-horizon) dynamic persuasion problem
between a sender and a receiver, both of whom are long-lived. At each
time $t$, there is a publicly-observable state $s_t$ and a
payoff-relevant quantity $\theta_t$, which is observed only by the
sender, and whose distribution depends on the current state (and
is independent of other quantities). Based on the observation of
$\theta_t$ the sender recommends an action to the receiver (which may
or may not be followed). The publicly-observable state then updates to
$s_{t+1}$ according to a transition kernel that depends on the current
state $s_t$, the quantity $\theta_t$ and the action chosen by the
receiver. Both the sender and the receiver seek to maximize their
total expected payoffs.

Previous work on this topic, with few exceptions, has focused on
myopic receivers, motivated by settings in which the receiver is
short-lived. The difference here then is the focus on long-lived,
far-sighted receiver. In this setting, the paper first shows (via an
example) that the class of Markovian signaling schemes (whose
recommendations only depend on the current state $s_t$) is
insufficient for optimal persuasion, and the sender can do better
using a signaling scheme that takes into account the history of the
process. Due to the computational difficulties in working with general
history-dependent schemes, the paper then considers promise-form
signaling schemes, which make recommendations based not only on the
current state, but also on a (history-dependent) "promise", which is a
guarantee on the receiver's continuation payoffs. Essentially, the
promise succinctly summarizes the history, thereby reducing the
computational complexity to be polynomial in the size of set of
promises. The authors show that, upon imposing a honesty condition on
the promises across time, the class of promise-form signaling schemes
suffice for optimal persuasion. The authors also propose an
approximation scheme for computing approximately-persuasive
promise-form signaling schemes with good payoff guarantees, that is
polynomial in the approximation factor.

**Strengths:**

+ The paper considers an interesting variation of the sequential
  persuasion problem, allowing for far-sighted receivers. This makes
  the problem substantially more complex. Nevertheless, the paper
  identifies a class of relatively simple and approximately persuasive
  signaling schemes that nevertheless achieve optimal payoffs for the
  sender, and are furthermore computationally tractable.

+ The paper illustrates well the insufficiency of the class of
  Markovian signaling schemes, and furthermore (adapting existing
  results) shows that finding a constant-factor approximation within
  the class of Markovian signaling schemes is NP-hard.

+ The class of promise-form signaling schemes is fairly simple and
  easy to implement; furthermore, it seems approximately-optimal such
  schemes can be computed via solving an LP (repeatedly).

**Weaknesses:**

+ While the class of promise-form signaling schemes is interesting,
  there is a significant line of work in economics that studies the
  use of promises in repeated games with incomplete information. The
  paper does not cite those papers, nor does it place its
  contributions within that context. A particularly relevant paper in
  this line is Abreu, Pierce and Stacchetti (Econometrica, 1990),
  whose results imply the sufficiency of the class of "promise-form"
  strategies for discounted repeated games with imperfect monitoring.

+ Similarly, the paper would benefit from connecting with general
  literature on repeated games (with or without incomplete
  information). For instance, the insufficiency of Markov signaling
  schemes is very much in the same vein as the inefficiency of Markov
  perfect equilibria in, say, repeated prisoner's dilemma to sustain
  cooperation. With far-sighted receivers, it is not surprising that
  Markov signaling schemes are not optimal for the sender.

+ $\epsilon$-persuasiveness: In the analysis of history-dependent (or
  promise-form) signaling schemes, the authors relax the
  persuasiveness requirement to $\epsilon$-persuasiveness. There is a
  subtle issue in interpreting this relaxation. To explain, a natural
  relaxation would be that the receiver's expected continuation payoff
  from following the recommendation is at most $\epsilon$ worse than
  choosing any other action, *after* receiving the recommendation.
  Specifically, the expectation taken here would be with respect to
  the posterior belief after receiving the recommendation. However,
  the condition in Definition 1 requires something different; it
  states that the receiver's expected payoff from following a
  recommendation, *multiplied* by the probability of receiving that
  recommendation, should be at most $\epsilon$ worse. In particular,
  there is an extra factor equaling the probability of recommending a
  particular action.

  While this may seem like a minor technical issue, this has
  substantial implication on the assumption that the receiver would
  adopt such a recommendation. For instance, this suggests that as
  long as the probability of recommending an action is small, the
  sender can recommend an action that can yield substantially lower
  continuation payoff for the receiver, and still expect the receiver
  to accept the recommendation. This seems to be a very strong
  assumption on the receiver's behavior, that does not align with the
  assumption that the receivers are (approximately) Bayesian.
  Moreover, with such a strong assumption, it is no longer clear if
  $OPT$ is the right benchmark for comparison.

  A potential fix to this issue would be to impose the relaxation on the
  conditional expectation, i.e., to replace the $\epsilon$ term in the
  definition with $\epsilon \sum_{\theta}
  \mu_h(\theta|s_h)\phi_\tau(a|\theta)$. However, it is not clear if
  the later approximation results continue to apply with this change.

+ Finally, while the paper makes sound and rigorous technical
  contribution, there is not enough discussion motivating the specific
  model being studied. For instance, there is no discussion of the
  motivation behind far-sightedness assumption; the myopic behavior of
  the receivers in previous work is frequently motivated by assuming a
  series of short-lived receivers. In particular, are there any
  specific applications where a single sender and a single receiver
  interact in the manner studied? (I think this is especially useful
  given the somewhat complicated form of the signaling scheme
  proposed.) Some discussion here would benefit the paper by grounding
  the theoretical results.

**Questions:**

+ Do the approximation results continue to hold if the relaxation of
  the persuasiveness constraint is imposed on the conditional
  expectation?

+ With the current definition of $\epsilon$-persuasiveness, it may be
  possible to design mechanisms that achieve payoffs substantially
  better than $OPT$. Are there any guarantees on how small (or large)
  this difference can be?

**Limitations:**

The assumptions are stated clearly. Some discussion of the limitations
induced by relaxing the persuasiveness/honesty requirements would be
helpful.

---

> ### Author Rebuttal · Authors · 2023-08-09
>
> We want to thank the Reviewer for their thoughtful comments and for mentioning pieces of related literature that we will reference and discuss in a final version of the paper. We provide below detailed answers to their questions.
>
> - *“Do the approximation results continue to hold if the relaxation of the persuasiveness constraint is imposed on the conditional expectation?”*
>
> Is the reviewer referring to the fact that the relaxation could be normalized by the probability with which the action is recommended? In this case, we believe that our techniques can be extended without any major difficulties changing the definition of $\epsilon$-persuasion and modifying the proofs accordingly. Let us know if you need further technical details.
>
> - *“With the current definition of $\epsilon$-persuasiveness, it may be possible to design mechanisms that achieve payoffs substantially better than $OPT$. Are there any guarantees on how small (or large) this difference can be?”*
>
> In general this difference can be arbitrarily large. Let us take a simple example with a single node, a single state, and two actions $a_1$ and $a_2$. The first action is $\epsilon$ better for the receiver with respect to the second one. The sender’s utility is $1$ if the receiver plays $a_2$ and $0$ if the receiver plays $a_1$. The optimal persuasive mechanism recommends $a_1$ and it has sender’s utility $0$. The optimal $\epsilon$-persuasive mechanism recommends $a_2$ and has sender’s utility $1$.
>
> This is an edge case. In non-degenerate settings the utility of the optimal persuasive and $\epsilon$-persuasive mechanisms are close.
>
> Moreover, we want to remark that this is standard in any setting with relaxed IC constraints, like $\epsilon$-Nash equilibria, $\epsilon$-Stackelberg, etc.

---

> > ### Comment · Reviewer_bcaS · 2023-08-10
> >
> > I was referring to the point that the actual persuasiveness constraint is on the conditional expectation, something along the lines of $\sum_{\omega} \phi(\omega | a) u(\omega, a) \geq \sum_{\omega} \phi(\omega | a) u(\omega, a')$. Here $\phi(\omega|a)$ is the conditional belief after receiving signal $a$. (I am using simplified notation here but hopefully it is still clear.)
> >
> > If this is relaxed to $\sum_{\omega} \phi(\omega | a) u(\omega, a) \geq \sum_{\omega} \phi(\omega | a) u(\omega, a') - \epsilon$, then this is a perfectly fine model of receiver behavior where one posits that any $\epsilon$-optimal action will be accepted by the receiver.
> >
> > However, currently the persuasiveness constraint is first modified to a unconditional form
> > $\sum_{\omega} \phi(\omega , a) u(\omega, a) \geq \sum_{\omega} \phi(\omega , a) u(\omega, a') $, where $\phi(\omega,a)$ is the joint distribution of state and signal. (This is common trick in exact static settings to get to an LP formulation.) Then, this modified constraint is relaxed to an approximate version:
> > $\sum_{\omega} \phi(\omega , a) u(\omega, a) \geq \sum_{\omega} \phi(\omega , a) u(\omega, a')- \delta$.
> >
> > However, if one goes back to the original (meaningful) persuasiveness constraint, this two-step relaxation amounts to modifying the original persuasiveness constraint as follows:
> > $\sum_{\omega} \phi(\omega |a) u(\omega, a) \geq \sum_{\omega} \phi(\omega | a) u(\omega, a') - \frac{\delta}{\phi(a)}$, where $\phi(a)$ is the unconditional probability of sending signal $a$.
> >
> > Now, this constraint posits a behavioral assumption on the receiver that they will be willing to follow the recommendation as along as it is approximately persuasive, where the approximation can depend also on the probability with which each signal is sent. In particular, as along as a signal is sent with small enough probability (i.e., as long as $\phi(a) \ll 1$), the receiver will gladly adopt the signal. Moreover, the behavior of the receiver changes based on the signal mechanism chosen by the sender. From a behavioral perspective, this situation seems very suspect.
> >
> > My question was whether the results in the paper continue to hold if we adopt the (more appropriate) approximation of  $\sum_{\omega} \phi(\omega | a) u(\omega, a) \geq \sum_{\omega} \phi(\omega | a) u(\omega, a') - \epsilon$.

---

> > > ### Author Response · Authors · 2023-08-16
> > >
> > > In the following we use *unnormalized* for describing the current definition of $\epsilon$-persuasiveness and with *normalized* the one proposed by the reviewer.
> > >
> > > We thank the reviewer for carefully clarifying this point. We decided to consider the unnormalized version of the $\epsilon$-persuasiveness since it is the standard way of defining approximate IC constraints in similar lines of work (e.g [Farina, Gabriele, et al. “Simple uncoupled no-regret learning dynamics for extensive-form correlated equilibrium”]). However we are now persuaded that the normalized version of the $\epsilon$-persuasive constraint is more appropriate for our work in which Bayesian rationality of the agents is a central concept and we agree that this makes the definition more sound from a behavioral perspective.
> > >
> > > Moreover, changing the definition to the normalized version comes at almost no cost. Indeed, we never directly use the definition of $\epsilon$-persuasiveness in the algorithmic part of the paper (Sections 5 and 6) . We use the definition of $\epsilon$-persuasiveness only in the proof of Lemma 3. In particular, we only use the fact that $\eta$-honest promise form signaling schemes are $H \eta$-persuasive (unnormalized definition) which is proved in Lemma 3. However the proof of Lemma 3 can be easily modified to show that $\eta$-honest promise form signaling schemes are $H \eta$-persuasive (normalized definition). Specifically, we only need to maintain the term $\sum_{\theta\in\Theta}\mu_h(\theta|s_h)\varphi_h(a| s_h,\iota_\tau^\sigma,\theta)$ in front of the $\eta(H-h)$ term in the last Equation after line 574. Notice that here we used an equivalent definition of normalized $\epsilon$-persuasiveness. In particular, using your notation, $\sum_\omega \phi(\omega|a) u(\omega,a) \ge \sum_\omega \phi(\omega|a) u(\omega,a’) - \epsilon$ is equivalent to $\sum_\omega \mu_\omega \phi(a|\omega) u(\omega,a) \ge  \sum_\omega \mu_\omega \phi(a|\omega) u(\omega,a’) -\epsilon \sum_\omega \mu_\omega \phi(a|\omega)$.
> > >
> > > This shows that all the results of our paper still apply to the new normalized version of approximate persuasiveness constraints. We will implement these changes in the final version of the paper. We are very thankful to the reviewer for the discussion which we think greatly improved our paper.

---

> > > > ### Comment · Reviewer_bcaS · 2023-08-17
> > > >
> > > > I acknowledge the response, and am glad to hear my comments were helpful in improving the paper.

---

### Author Rebuttal · Authors · 2023-08-09

We want to thank the reviewers for their useful feedback and for praising the technical contribution of our work. We will take their suggestions in great consideration to improve the final version of the paper in terms of discussion of additional related works, considered assumptions, and real-world applications.

Before providing detailed replies to reviewers’ questions below, we want to uphold two important choices in our problem formulation.

While reviewers rightly noted that the known-model assumption is restrictive, we think that solving the planning setting already required to overcome significant technical hurdles, which left no space for additional contributions. Moreover, we believe that the planning problem is a natural preliminary step towards addressing the corresponding learning problem, in which the model is (at least partially) unknown to the agents. The latter problem is a nice direction for future research on this topic.

Finally, we will make an additional effort to explain why the farsighted receiver assumption matters in this setting. First, it complements previous works considering myopic receivers in MDPs, which is sometimes unrealistic in real-world applications, such as ride-sharing platforms where the same driver typically interacts with the platform multiple times. Secondly, it better aligns this line of research on information design in MDP settings with the classical MDP literature, which crucially assumes farsighted decision makers.

---

### Decision · Program_Chairs · 2023-09-21

**Decision:**

Accept (poster)

**Comment:**

I've skimmed the paper, read the reviews and the rebuttals, and discussed the paper with the reviewers and the AC. I think this is a good paper which should be published.

However, there are two major caveats:

- The notion of $\epsilon$-persuasiveness in the submitted paper is not well-motivated. However, the reviewers are convinced the results carry over to a stronger and well-motivated notion (as per the reviewer-authors discussion).

- A blatant (but easily fixable) flaw in presentation. What is the receiver's behavioral against $\epsilon$-persuasive policy? The definition (L107) says "exactly rational", whereas the technical developments (e.g., equations in L227-232) are consistent with "follow the policy".

While it is unusual to allow such substantive changes from the submission to the camera-ready version, we decided to give the authors the benefit of a doubt. I emphasize that **these changes are necessary for the paper to be accepted**
.

**Minor issue:** The "stopping assumption" (L115-117) states that the receiver stops issuing recommendations once any of them is not followed . The following should be clarified:

- It is not an "assumption" per se, but a design choice, and a fairly reasonable one.

- It is w.l.o.g. under 0-persuasiveness, and therefore does not affect the OPT. However, it might worsen the performance of $\epsilon$-persuasive policies, for all we know.

- As the main results (Thms 4 and 5) restrict the policy class anyway, this design choice can be interpreted as a part of this restriction.